# CLIP-Guided Reinforcement Learning for Open-Vocabulary Tasks

## Abstract

Open-vocabulary ability is crucial for an agent designed to follow natural language instructions. In this paper, we focus on developing an open-vocabulary agent through reinforcement learning. We leverage the capability of CLIP to segment the target object specified in language instructions from the image observations. The resulting confidence map replaces the text instruction as input to the agent's policy, grounding the natural language into the visual information. Compared to the giant embedding space of natural language, the two-dimensional confidence map provides a more accessible unified representation for neural networks. When faced with instructions containing unseen objects, the agent converts textual descriptions into comprehensible confidence maps as input, enabling it to accomplish open-vocabulary tasks. Additionally, we introduce an intrinsic reward function based on the confidence map to more effectively guide the agent towards the target objects. Our single-task experiments demonstrate that our intrinsic reward significantly improves performance. In multi-task experiments, through testing on tasks out of the training set, we show that the agent, when provided with confidence maps as input, possesses open-vocabulary capabilities.

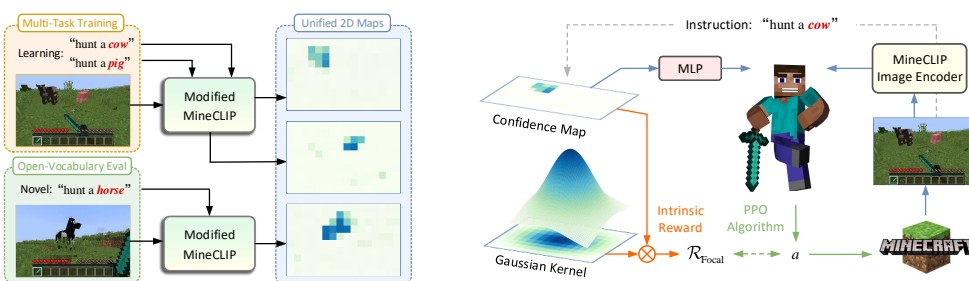

Figure 1: Overview of **CLIP**-guided **O**pen-vocabulary **P**olicy **L**earning (**COPL**). (*left*) COPL tackles open-vocabulary tasks by mapping the novel object into a comprehensible unified 2D confidence map, relying on our modified MineCLIP. (*right*) The agent takes as input the image observation and the confidence map of the target specified by the instruction. We train the agent by PPO with our proposed focal reward derived from the confidence map to guide the agent toward the target.

## 1 Introduction

In the field of artificial intelligence, the ability of agents to understand and follow natural language instructions in an open-ended manner is crucial (Brohan et al., 2022; 2023; Chen et al., 2023; Shah et al., 2023). However, the scope of training content is always finite. Open-vocabulary tasks, where the agent is instructed to interact with diverse objects, *beyond the training scope*, from the vast realm of human vocabulary, represent a pivotal step towards creating general AI systems capable of adapting to a wide range of real-world scenarios (Chen et al., 2023; Stone et al., 2023). As a popular open-ended 3D game, Minecraft serves as an ideal testbed for learning and evaluating open-vocabulary ability. At its core, Minecraft offers procedurally generated worlds with unlimited size and a large variety of tasks ranging from navigation and combat to building and survival (Fan et al., 2022; Wang et al., 2023b; Yuan et al., 2023; Wang et al., 2023a; Zhu et al., 2023). Compared with canonical game environments such as Go (Silver et al., 2016), Atari (Mnih et al., 2013), and

StarCraft (Vinyals et al., 2019), Minecraft mirrors the complexity of real-world challenges and offers a wide range of objects and tasks with natural language instructions.

To equip an agent with open-vocabulary ability, the integration of a vision-language model (VLM) is promising (Wu et al., 2023). A VLM aligns images and language vocabularies into the same feature space, bridging the gap between visual observations and natural language instructions. Therefore, it has the capability to ground the agent's unseen text, *e.g.*, names of novel objects, into visual images, enabling the agent to comprehend instructions not encountered during training. Thanks to MineCLIP (Fan et al., 2022), a VLM pre-trained on Internet-scale Minecraft videos from YouTube, developing an open-vocabulary agent in Minecraft has become more accessible. Initially, MineCLIP was merely used as a tool to measure the similarity between a sequence of visual observations and the instruction, serving as an intrinsic reward for reinforcement learning (Fan et al., 2022). Recent advancement has taken a further step to exploit the capabilities of MineCLIP. STEVE-1 (Lifshitz et al., 2023) converts natural language instructions into the embedding space via the MineCLIP encoder and leverages this embedding to guide VPT, a foundation model of Minecraft behaviors (Baker et al., 2022). This innovation steps towards open-vocabulary agents, as it enables the agent to comprehend diverse and free-form language instructions.

While MineCLIP has already demonstrated its power through STEVE-1, its capabilities are yet to be fully explored. As a model fine-tuned from CLIP (Radford et al., 2021), MineCLIP inherits most characteristics of CLIP. Recent works in computer vision have extensively adopted CLIP as a foundation model for open-vocabulary object detection (Gu et al., 2021; Kuo et al., 2022; Zang et al., 2022) and open-vocabulary segmentation (Ding et al., 2022; Rao et al., 2022; Liang et al., 2023), leveraging its rich knowledge. Moreover, CLIP even exhibits remarkable segmentation capabilities and explainability without fine-tuning (Zhou et al., 2022; Li et al., 2023). These findings indicate that MineCLIP would also possess the capability to locate and segment the target object specified in the language instruction from the image observation in Minecraft.

The ability of MineCLIP to perform segmentation provides three key inspirations for enhancing agent learning in Minecraft. Firstly, taking as input the location information of the target object would facilitate training and improve performance, as it offers a direct means of grounding natural language into the image. Practical research in robotics has proven that models with such location input show superior performance compared to text input (Stone et al., 2023). Secondly and most significantly, the segmentation is open-vocabulary. Therefore, when the agent receives instructions containing novel objects not encountered in the training phase, the segmentation remains effective. Lastly, it is noticeable that the intrinsic reward calculated by MineCLIP (Fan et al., 2022) has one limitation: it is insensitive to the distance to the target object (Radford et al., 2021; Cai et al., 2023). Fortunately, with the segmentation result, the pixel area of the target object can serve as a surrogate for distance, providing more information to calculate a better intrinsic reward.

In this paper, we propose a **CLIP-guided Open-vocabulary Policy Learning** method, namely **COPL**. We generate a confidence map of the target object specified in the language instruction via our modified MineCLIP. We extend MineCLIP with modifications inspired by MaskCLIP (Zhou et al., 2022) so that it can segment the specified object from the image. As illustrated in Figure 1 (*left*), our approach can convert instructions into unified two-dimensional maps. To leverage this result, we first design an intrinsic reward that takes into account the pixel area and location of the target object in the image observation. By doing so, we address the deficiency of the original MineCLIP reward (Fan et al., 2022). Furthermore, we integrate the resulting confidence map into the policy input, instead of text input or other task indicators, as illustrated in Figure 1 (*right*). Based on this adjustment, our agent is able to handle open-vocabulary tasks through multi-task reinforcement learning on *only a limited set of instructions*.

We evaluate COPL on basic skill learning and open-vocabulary generalization in Minecraft. Firstly, we conduct a group of single-task experiments to show that our refined intrinsic reward significantly outperforms the MineCLIP reward in enabling the agent to successfully acquire various challenging basic skills. Then we extend our evaluation to instruction-following scenarios, where we train the agent with a set of instructions. In our test, the agent exhibits the capacity to execute instructions involving previously unseen targets, effectively demonstrating its open-vocabulary ability. Though we implement and evaluate COPL in Minecraft, we believe our method is extendable to other similar open-world environments and draws insights into the integration of VLM and reinforcement learning for training open-vocabulary agents.

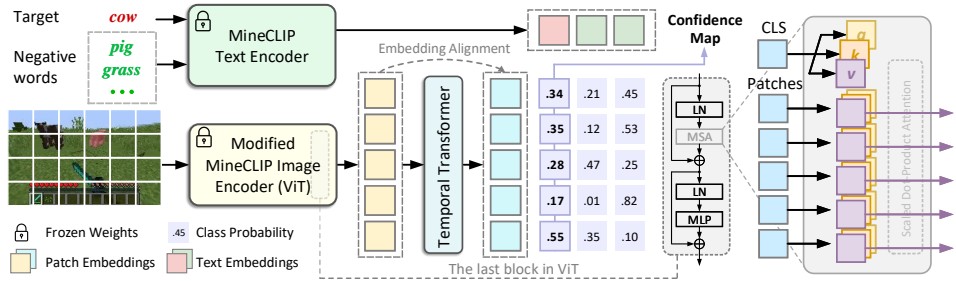

Figure 2: Process of segmentation via MineCLIP. The modified MineCLIP image encoder takes as input the image and outputs patch embeddings, which are subsequently processed by the temporal transformer to guarantee embedding alignment. The MineCLIP text encoder encodes the target name along with a list of negative words. The probability of the target's presence on each patch is calculated based on the similarities between patch embeddings and text embeddings.

## 2 PRELIMINARY

**Problem Statement.** In this paper, by open-vocabulary task, we mean that the agent is instructed to interact with diverse objects beyond the training scope. More specifically, we focus on *object-centric tasks* and the *open-vocabulary ability over target objects*. To formalize, we denote the set of objects with which the agent learns to interact during the training phase as $C_t$, and the set of objects with which the agent is required to interact during the execution phase as $C_e$. To test the open-vocabulary ability of the agent, $C_e$ consists of objects that are not in $C_t$. For example, during training, the agent learns to accomplish language instructions "*hunt a cow*" and "*hunt a sheep*". However, during execution, it will encounter instructions like "*hunt a horse*" or "*hunt a chicken*", where neither "*horse*" nor "*chicken*" appears in the instructions during training. Note that we do not consider open-vocabulary ability concerning actions (we leave it as future work). Therefore, instructions during execution should have the *same behavior patterns* as those learned in training. For instance, when training with "*hunt sth.*" and "*harvest sth.*", testing with "*explore the world*" is not considered.

Given that we choose reinforcement learning to train the agent, a similar problem is zero-shot generalization in reinforcement learning (Kirk et al., 2023). The difference between zero-shot generalization and open-vocabulary tasks is that the former focuses on the agent's adaptability to unseen contexts, including environments with different dynamics or backgrounds, while the latter cares about how to generalize the learned skill to unseen target objects specified by instructions. Both problems demand adaptability and generalization but differ in the range of scenarios they address.

**MineCLIP for Minecraft RL.** MineCLIP is a vision-language model pre-trained on Internet-scale Minecraft videos from YouTube (Fan et al., 2022). This model learns the alignment between video clips (consisting of 16 frames) and natural language. Similar to CLIP (Radford et al., 2021), MineCLIP adopts a ViT (Dosovitskiy et al., 2020) as the image encoder and a GPT (Radford et al., 2019) as the text encoder. The main difference between MineCLIP and CLIP is that MineCLIP takes as input a sequence of 16 images. Therefore, MineCLIP incorporates an additional module to aggregate the 16 embeddings generated by the image encoder. The proposed two mechanisms include a temporal transformer (MineCLIP[attn]) and direct average pooling (MineCLIP[avg]). In this paper, we choose the former as our base model due to its better performance in Programmatic tasks compared to the latter (Fan et al., 2022). For reinforcement learning in Minecraft, MineCLIP provides an intrinsic reward function $\mathcal{R}_i : \mathcal{G} \times \mathcal{S}^{16} \to \mathbb{R}$, representing the similarity between the observation sequence of the previous 16 steps $[s_{t-15}, \cdots, s_{t-1}, s_t]$ and the task prompt $g$.

## 3 METHOD

In this section, we detail the implementation of our COPL method addressing open-vocabulary tasks in Minecraft. We introduce the modification to MineCLIP (Fan et al., 2022) and the process of segmenting the target object specified by the language instruction (Section 3.1). This process yields a confidence map, where each element represents the probability of the specified target's presence. Based on this confidence map, we present a *simple but effective* intrinsic reward to guide the agent

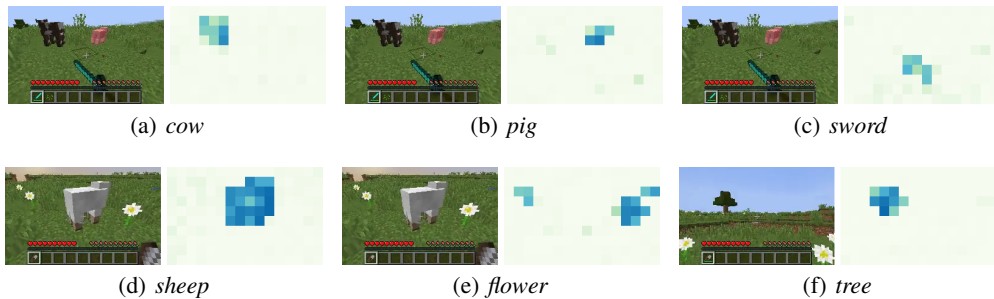

Figure 3: Segmentation instances for targets: (a) *cow*, (b) *pig*, (c) *sword*, (d) *sheep*, (e) *flower*, and (f) *tree*. *Left*: The original image. *Right*: The confidence map of the target. The darker blue the patch, the higher the probability of the target's presence on it.

toward the target, facilitating the learning of basic skills during training (Section 3.2). We integrate the confidence map, which contains essential spatial information of the specified target, into the policy as input (Section 3.3). This integration equips the agent with open-vocabulary ability by grounding the novel object into a comprehensible input, *i.e.*, the confidence map.

## 3.1 SEGMENTATION VIA MINECLIP

Prior to segmentation, we must extract the correct target that the agent needs to interact with from the provided language instruction. Consider an example instruction: "*hunt a cow in plains with a diamond sword*". In this case, it is "*cow*" that should be extracted from the instruction, rather than "*plains*" or "*diamond sword*", for the following segmentation. This can be easily done by large language models (LLMs). Details can be found in Appendix A.1.

In the standard CLIP (Radford et al., 2021), the image encoder, a ResNet (He et al., 2016) or ViT (Dosovitskiy et al., 2020), aggregates the visual features from all spatial locations through attention pooling. Recent works (Zhou et al., 2022; Li et al., 2023) reveal that these features on each spatial location contain rich local information so that they can be used to perform zero-shot pixel-level predictions. In brief, the cosine similarities between these features and the outputs of the CLIP text encoder are also valid and informative. Concretely, MaskCLIP (Zhou et al., 2022) makes use of the value-embedding of each spatial location in the last attention module, while CLIPSurgery (Li et al., 2023) studies the feature of each spatial location in the final output and introduces an additional path. Inspired by MaskCLIP, we make adaptations to MineCLIP architecture to generate a confidence map for a specified target *without fine-tuning*.

To begin, we introduce the modification to the vision pathway of MineCLIP. We make changes to extract dense features from the last block of ViT. As illustrated in the rightmost part of Figure 2, the scaled dot-product attention in multi-head attention (Vaswani et al., 2017) module is removed, while the *value-embedding transformation* is retained. Then the transformed embeddings excluding that of CLS token are fed into the remaining modules within the ViT to obtain the final embedding of each patch. In this way, these patch embeddings share the same space as the original ViT output. As shown in Figure 2, the modified image encoder outputs patch embeddings instead of image embedding. However, these embeddings are not yet aligned with the embedding space of MineCLIP. In MineCLIP, the image encoder is followed by a temporal transformer that aggregates the embeddings of 16 images. Therefore, these patch embeddings also need to pass through the temporal transformer to guarantee alignment. Notably, these embeddings do not form a temporal sequence together as the input of the transformer. Instead, each patch embedding is individually processed by the temporal transformer, treated as a sequence of length 1. In this way, we obtain patch embeddings in the MineCLIP embedding space.

In the language pathway, no modification is made to the MineCLIP text encoder. The target name is encoded using the text encoder, along with a list of negative words. We construct a negative word list containing objects that frequently appear in Minecraft. For a detailed description of the word list, please refer to Appendix A.2. Given the patch embeddings encoded through the modified image encoder and the temporal transformer in the same embedding space of MineCLIP, we can calculate cosine similarities between patch embeddings and text embeddings, following the same approach as

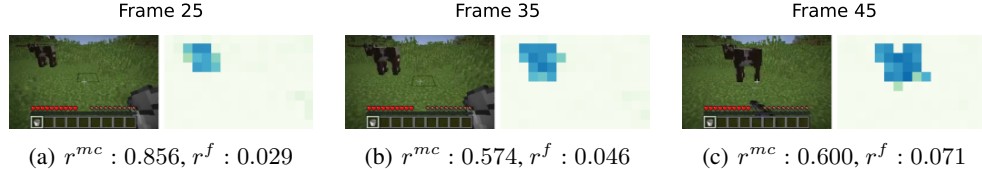

| Frame 25 | Frame 35 | Frame 45 |
|---|---|---|

(a) $r^{mc} : 0.856, r^f : 0.029$     (b) $r^{mc} : 0.574, r^f : 0.046$     (c) $r^{mc} : 0.600, r^f : 0.071$

Figure 4: Comparison between MineCLIP reward $r^{mc}$ and focal reward $r^f$ at Frame 25, 35, and 45, in one episode of the task "*milk a cow*". From (a) to (c), our focal reward consistently increases as the agent approaches the target *cow*, while the MineCLIP reward varies in an uncorrelated way.

CLIP. Subsequently, we use softmax with the same temperature used in MineCLIP to determine the probabilities of objects' presence on each patch. Finally, we extract and reshape the probabilities of the target object to form the confidence map. The resulting confidence map consists of the same number of elements as the patches, with each element representing the probability of the target's presence on the corresponding patch. Examples of the confidence maps are shown in Figure 3.

## 3.2 FOCAL REWARD

As noted in Cai et al. (2023), the MineCLIP reward, which relies on the similarity between the agent's preceding image observations and the provided instruction, is uncorrelated with the distance between the agent and the target. This phenomenon is demonstrated in Figure 4, where the MineCLIP reward does not consistently increase as the agent gets closer to the target. Consequently, in practice, the agent trained with the MineCLIP reward tends to "stare at" the target at a distance, rather than approaching it. This tendency obstructs the agent from learning some hard-exploration skills, particularly those that require multiple times of interactions with the targets, such as hunting.

Fortunately, the confidence map of the target contains rich spatial information that can mitigate the limitations of the original MineCLIP reward. The area occupied by the target in the image can serve as a *proxy* for estimating the distance to the target, based on the principle that the closer the target is to the agent, the larger its area in the image and vice versa. Therefore, a reward proportional to the area of the target would guide the agent towards the target effectively. Additionally, we argue that the agent should be encouraged to aim at the target, *i.e.*, adjust the perspective to center the target in the field of view. This would help the agent further stabilize its orientation and increase the chance of interacting with the target when it is close enough. In Minecraft, interaction can only occur when the cursor in the center of the agent view aligns with the target. Moreover, when multiple target objects are present in the view, the agent should learn to focus on a single target rather than attempting to keep all of them in view. This could also be interpreted in a more general way, such as humans usually place the target at the center of the visual field for better perception and interaction.

Based on these principles, we introduce an intrinsic reward function named *focal* reward. At each time step $t$, it is computed as the mean of the Hadamard product between the target confidence map $m_t^c$, and a Gaussian kernel denoted as $m^k$:

$$r_t^f = \text{mean}\left(m_t^c \circ m^k\right). \tag{1}$$

Here, $m_t^c$ and $m^k$ share the same dimensions with height $H$ and width $W$. Each element of the Gaussian kernel is defined as:

$$m_{i,j}^k = \exp\left(-\frac{(i-\mu_1)^2}{2\sigma_1^2} - \frac{(j-\mu_2)^2}{2\sigma_2^2}\right), i \in \{1, ..., H\}, j \in \{1, ..., W\}, \tag{2}$$

where $\mu_1 = (H+1)/2$, $\sigma_1 = H/3$, $\mu_2 = (W+1)/2$, and $\sigma_2 = W/3$. This reward function is designed to be directly proportional to the area occupied by the target and inversely proportional to the distance between the target patches and the center of the view. As illustrated in Figure 4, when the agent approaches the target *cow*, the region of high confidence becomes larger and closer to the center, and consequently, our focal reward increases consistently.

The confidence map generated from the modified MineCLIP may sometimes contain noisy activation (Zhou et al., 2022; Li et al., 2023). Therefore, we process the raw confidence map to enhance its quality before using it to compute the intrinsic reward. Firstly, we set the value corresponding to

the patch where a word from the negative word list has the highest probability instead of the target to zero. This operation diminishes the influence of noisy activation on non-target patches. Secondly, we set values in the confidence map lower than a threshold $\tau = 0.2$ to zero, while those higher than this threshold are set to one, so as to amplify the distinction between patches corresponding to the target and those unrelated to it. We ablate the Gaussian kernel and denoising process in Section 4.1.

### 3.3 OPEN-VOCABULARY POLICY LEARNING

To train an instruction-following agent, the conventional practice involves directly taking the natural language instruction as input into the policy network (Khandelwal et al., 2022; Mu et al., 2022; Du et al., 2023). These instructions are typically encoded using a recurrent network or a language model such as BERT (Kenton & Toutanova, 2019). In contrast, we extract the target object from the instruction using ChatGPT (OpenAI, 2022) and subsequently convert it into a two-dimensional matrix, *i.e.*, the confidence map. Our underlying assumption is that this two-dimensional spatial representation offers more intuitive and accessible information for the policy network compared to the intricate space of language embeddings. When facing an instruction containing the name of an unseen target object during execution, our method grounds this novel text into the two-dimensional map, rendering it comprehensible to the policy network. As a result, the agent can follow the guidance of the confidence map, navigate towards the novel target object, and finally interact with it.

In our implementation, we adopt the network architecture of MineAgent (Fan et al., 2022), which uses the MineCLIP image encoder to process image observations and MLPs to encode other information such as pose. We introduce an additional branch to encode the confidence map and fuse these features through concatenation. The policy network takes this fused multi-modality feature as input and outputs action distribution. Details regarding the policy network's architecture are available in Appendix B.2. We use PPO (Schulman et al., 2017) as the base RL algorithm and train the agent with reward $r_t = r_t^{env} + \lambda r_t^f$, where $r^{env}$ denotes the environmental reward and $\lambda$ is a hyperparameter controlling the weight of the focal reward. According to the empirical results in Appendix B.4, we simply set $\lambda = 5$ for all experiments in the paper as we do not want to bother tuning this hyperparameter. We employ the multi-task reinforcement learning paradigm, where the agent is trained to finish tasks in a predefined instruction set. Unlike typical multi-task reinforcement learning, our agent's learning objective is to not only master the training tasks but also to understand the mapping between the confidence map and the target object within the image observation in order to perform zero-shot transfer to novel instructions.

## 4 EXPERIMENTS

We conduct experiments in MineDojo (Fan et al., 2022), a Minecraft simulator that offers diverse open-ended tasks. We perform single-task experiments to evaluate the effectiveness of our proposed focal reward. Then we extend our evaluation to multi-task experiments, and most importantly, open-vocabulary tasks. Details about Minecraft environments and RL hyperparameters in our experiments are described in Appendix B.1 and Appendix B.4, respectively.

### 4.1 SINGLE-TASK EXPERIMENTS

Our single-task evaluation consists of tasks learning four challenging basic skills: *hunt a cow*, *hunt a sheep*, *hunt a pig*, and *hunt a chicken*. In each task, the agent spawns in `plains` biome alongside several animals. The agent will receive a reward from the environment if it successfully kills the target animal. The difficulty of these basic skills lies in that animals, once attacked, will attempt to flee, requiring the agent to keep chasing and attacking the target animal. More details about Minecraft task settings are available in Appendix B.3.1.

**Evaluation.** We compare our focal reward with the following baselines: (1) MineCLIP reward (Fan et al., 2022) based on the similarity between image observations and the instruction "*hunt a {animal} on plains with a diamond sword*"; (2) $ND_{CLIP}$ reward (Tam et al., 2022), an intrinsic reward for exploration that measures the novelty of observation's MineCLIP embedding; (3) Sparse reward, *i.e.*, training the agent with the environmental reward only. Results are reported in Figure 5. Each curve shows the mean success rate of four runs with different seeds and shaded regions indicate standard error (the same applies hereinafter). We can observe that only our focal reward leads to the mastery

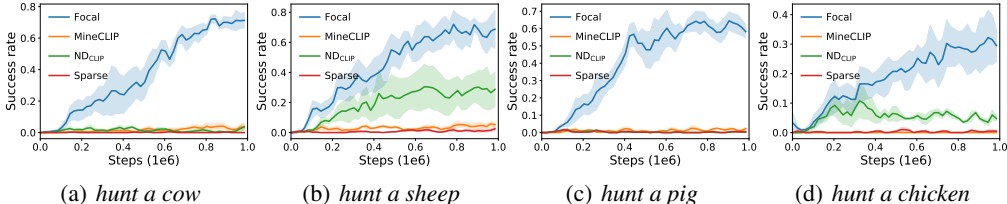

(a) *hunt a cow*    (b) *hunt a sheep*    (c) *hunt a pig*    (d) *hunt a chicken*

Figure 5: Learning curves of PPO with focal reward, MineCLIP reward, ND$_{CLIP}$ reward, and environmental sparse reward only, on four Minecraft tasks.

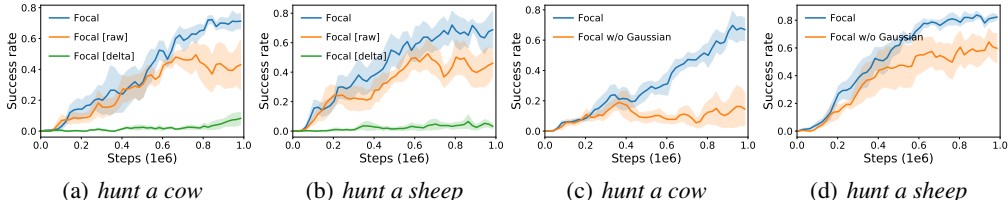

(a) *hunt a cow*    (b) *hunt a sheep*    (c) *hunt a cow*    (d) *hunt a sheep*

Figure 6: (a)(b) Learning curves of PPO with focal reward and its variants. (c)(d) Learning curves of PPO with focal reward and focal reward without Gaussian kernel.

of all four skills by guiding the agent to consistently approach the target. In contrast, the MineCLIP reward fails because it cannot capture the distance between the agent and the target, offering limited benefit to these tasks. The failure of ND$_{CLIP}$ reward suggests that exploration provides minimal assistance in learning these challenging skills due to the huge observation space of Minecraft.

**Variants and Ablation.** To further investigate our focal reward, we compare it with two variants: Focal[raw], which uses the raw confidence map without denoising to compute the intrinsic reward, and Focal[delta], defined as $r_t^\delta = r_t^f - r_{t-1}^f$. The results in Figures 6(a) and 6(b) demonstrate that our denoising process improves the effectiveness of the focal reward. We suppose that the poor performance of Focal[delta] may be linked to its sensitivity to segmentation noise, as it relies on differences in focal reward between two steps, making it susceptible to minor fluctuations in segmentation. In addition, we test the effectiveness of the Gaussian kernel, as presented in Figures 6(c) and 6(d). We modify the environment settings to ensure that there are two target animals. The results prove the significance of the Gaussian kernel. Without this kernel, the reward may guide the agent to include both target animals in the view to acquire a high reward, hindering it from approaching either of them. In contrast, our focal reward addresses this problem by providing more reward in the center, thereby encouraging the agent to focus on a single target.

### 4.2 MULTI-TASK AND OPEN-VOCABULARY EXPERIMENTS

We conduct multi-task experiments to verify the effectiveness and open-vocabulary capability of COPL. Given that tasks in Minecraft require different behavior patterns, we design two task domains, the **hunt domain** and the **harvest domain**. The hunt domain consists of four instructions in `plains` biome: "*hunt a cow*", "*hunt a sheep*", "*hunt a pig*", and "*hunt a chicken*". These tasks share a common behavior pattern: repeatedly *approach the target, aim at it, and attack*. The harvest domain contains two instructions in `plains` biome, "*milk a cow*" and "*shear a sheep*", and two instructions in `flower_forest` biome, "*harvest a flower*" and "*harvest leaves*". Tasks in the harvest domain are individually easier than those in the hunt domain but demand disparate behavior patterns. For example, "*harvest a flower*" requires the *attack* action while the other tasks require the *use* action. More details about the task settings are available in Appendix B.3.3.

**Evaluation.** We compare COPL with two baselines: (1) EmbCLIP (Khandelwal et al., 2022), utilizing the target embedding provided by the MineCLIP text encoder as input; (2) One-Hot, a naive multi-task baseline, using a one-hot vector as the task indicator. All these methods are trained with the focal reward and the only difference is their target representations. In the hunt domain, as shown in Figure 7(a), COPL significantly outperforms other baselines, indicating that the confidence map provides a more accessible and informative target representation compared to the language embedding and one-hot vector, respectively. Notably, One-Hot surpasses EmbCLIP, suggesting that the intricate language embedding of the target may have a negative impact on multi-task learning. In

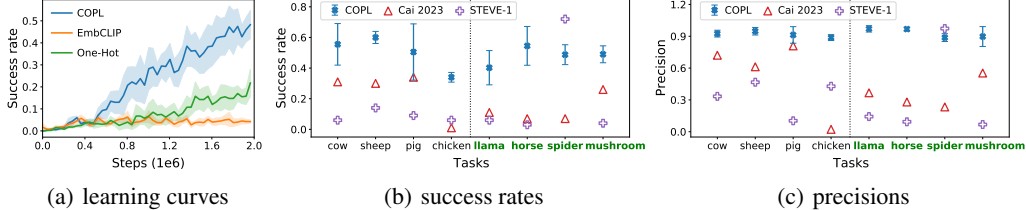

(a) learning curves      (b) success rates      (c) precisions

Figure 7: (a) Learning curves of COPL, EmbCLIP, and One-Hot in the hunt domain. (b) Success rates and (c) precisions of COPL, Cai et al. (2023), and STEVE-1 on each hunt task. Solid × marks and their error bars represent the mean and variance of COPL, respectively. Hollow marks denote the performance of a single model, so they do not have error bars. Novel tasks are **highlighted**.

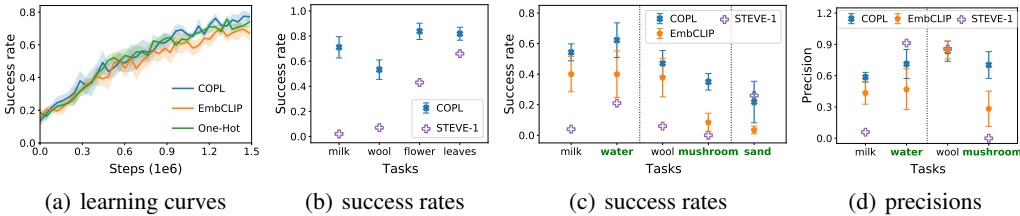

(a) learning curves    (b) success rates    (c) success rates    (d) precisions

Figure 8: (a) Learning curves of COPL, EmbCLIP, and One-Hot in the harvest domain. (b) Success rates of COPL and STEVE-1 on each training task. (c) Success rates and (d) precisions of COPL, EmbCLIP, and STEVE-1 on each test harvest task. Novel tasks are **highlighted**.

contrast, the harvest domain presents a different picture. As illustrated in Figure 8(a), all methods achieve similar performance. These results suggest that when tasks become easy enough, the impact of the target representation's complexity diminishes. These methods' learning curves on each task are available in Appendix B.5. We also benchmark COPL against two recent Minecraft basic skill models trained via imitation learning, Cai et al. (2023)[1] and STEVE-1 (Lifshitz et al., 2023). COPL outperforms both models significantly across all tasks, as shown in Figures 7(b) and 8(b).

**Open-Vocabulary Generalization.** Given that the two domains involve distinct behavior patterns, we conduct separate evaluations to assess the open-vocabulary ability of COPL models trained in the hunt domain and the harvest domain. Besides four learning instructions, we test the hunt domain model with four novel instructions in `plains` biome, "*hunt a llama*", "*hunt a horse*", "*hunt a spider*", and "*hunt a mushroom cow*". The results in Figure 7(b) show that COPL effectively transfers the learned skill to unseen targets, achieving high success rates. Additionally, we define precision as the number of correct kills on the specified target divided by the number of kills on any animal. The high precision, as reported in Figure 7(c), proves COPL's ability to distinguish the target from other animals, rather than indiscriminately attacking them. STEVE-1 shows poor performance across all hunt tasks except "*hunt a spider*". We suppose that its base model, VPT (Baker et al., 2022), possesses a strong prior on killing specific animals like spiders and heavily affects the behavior of STEVE-1 on following other hunting instructions. Cai et al. (2023) achieves relatively higher success rates on "*hunt a cow*", "*hunt a sheep*", and "*hunt a pig*" due to these tasks being in its training set. Its lower performance on other tasks indicates its limitations in open-vocabulary ability. "*Hunt a mushroom cow*" is an exception and we hypothesize that this is because the mushroom cow is similar to the cow in shape and texture.

Considering the diverse behavior patterns and tools used in the harvest domain, we test our harvest domain model using three groups of instructions: (1) "*milk a cow*" and "*harvest water*" in `river` biome, both requiring the agent to *use* an empty bucket; (2) "*shear a sheep*" and "*shear a mushroom cow*" in `plains` biome, both requiring the agent to *use* shears; (3) "*harvest sand*" in `river` biome, sharing a similar *attack* behavior with "*harvest a flower*" but equipped a unseen tool, a diamond shovel. Results are depicted in Figures 8(c) and 8(d). Precision here is defined as the number of times correctly harvesting the specified target divided by the number of times harvesting any target declared in the group's instructions. Our results reveal that although COPL and

---

[1] We do not evaluate the performance of Cai et al. (2023) in the harvest domain because the authors have not yet released the model trained for harvest tasks.

EmbCLIP show similar performance on training tasks, COPL exhibits advantages on novel tasks, achieving higher success rates and precisions compared to EmbCLIP. This indicates that better open-vocabulary ability emerges from converting language into a simple two-dimensional representation. STEVE-1 achieves a decent performance only on "*harvest sand*" due to its powerful digging skills.

## 5 RELATED WORK

**Minecraft Research.** Broadly, challenges in Minecraft can be categorized into high-level task planning and low-level skill learning. For high-level planning, where agents must make decisions on which skills to employ sequentially based on the given instruction, the field has converged towards leveraging the Large Language Model (LLM) (Nottingham et al., 2023; Wang et al., 2023b;a; Yuan et al., 2023; Zhu et al., 2023). Regarding learning low-level skills, the difficulty lies in the absence of well-defined dense reward and a vast variety of objects to interact with in Minecraft. Unlike the convergence in high-level planning approaches, two distinct routes have emerged in low-level learning. The first route, represented by MineCLIP (Fan et al., 2022), utilizes the reward derived from the alignment between text and video clip or other manually designed reward for reinforcement learning (Yuan et al., 2023). The second one follows the principles of VPT (Baker et al., 2022), where skills are acquired through imitation learning based on large-scale demonstration (Cai et al., 2023; Lifshitz et al., 2023). Our work falls in the scope of low-level skill learning with reinforcement learning.

**Instruction-Following RL.** Language has been widely explored in goal-conditioned reinforcement learning for its compositional structure (Luketina et al., 2019). This feature allows goal-conditioned policies to better capture the latent structure of the task space and generalize to unseen instructions that combine seen words (Oh et al., 2017; Chan et al., 2019; Jiang et al., 2019; Colas et al., 2020; Mirchandani et al., 2021). With the development of LLM and VLM, language also becomes a means of providing intrinsic rewards in reinforcement learning. The similarity or correlation between instructions and current states provides dense rewards to guide the agent's learning more effectively (Fan et al., 2022; Kwon et al., 2022; Mahmoudieh et al., 2022; Du et al., 2023). Our work stands out by enabling the policy to generalize to instructions that contain previously unseen targets.

**CLIP for Embodied AI.** CLIP (Radford et al., 2021) provides diverse usage for AI research. We categorize these applications into three areas: *encoding*, *retrieving* and *locating*. Encoding, the most common use of CLIP, leverages CLIP encoders to represent images and/or texts (Shridhar et al., 2022; Khandelwal et al., 2022; Majumdar et al., 2022). Our work also utilizes the MineCLIP image encoder to process raw image observations. Retrieving mostly involves navigation tasks, where CLIP assists in selecting the most matching image from a set based on the given instruction (Dorbala et al., 2022; Bucker et al., 2023; Chen et al., 2023; Shah et al., 2023). The most relevant usage to our work is locating, which applies methods like MaskCLIP (Zhou et al., 2022) or GradCAM (Selvaraju et al., 2017) on CLIP to determine the position of the specific object in images (Wang et al., 2022; Gadre et al., 2023; Zhang et al., 2023). Based on the object location, agents can conduct planning with a depth detector (Gadre et al., 2023) or imitation learning (Wang et al., 2022; Zhang et al., 2023). In contrast, our work focuses on training agents via reinforcement learning with information solely extracted from image observations, without any extra spatial information or demonstration.

## 6 CONCLUSION

In this paper, we propose COPL, a novel approach designed to address open-vocabulary tasks in Minecraft, leveraging the wealth of knowledge about Minecraft encoded in MineCLIP (Fan et al., 2022). Through comprehensive evaluations, we prove COPL's effectiveness in acquiring multiple basic skills and its open-vocabulary ability. Additionally, we demonstrate the advantages of training policies through reinforcement learning: the performance is not dependent on the quality and distribution of demonstration, allowing the trained policy to handle tasks that are challenging but less common in human-collected data, such as hunting animals (Baker et al., 2022). Furthermore, our work demonstrates the potential of integrating multimodal models, such as VLM, into reinforcement learning. Our method can be applied to other similar open-world environments by grounding natural language instructions into visual data and guiding the agent toward targets likewise. We hope COPL could contribute to the development of agents capable of understanding and responding to natural language instructions. Future work could focus on grounding language that describes actions and learning tasks requiring more complicated manipulation.

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

# A    SEGMENTATION DETAILS

## A.1    EXTRACTING TARGETS VIA LLM

In order to extract the target name from a natural language instruction, we resort to ChatGPT (OpenAI, 2022). We input the following prompt:

```
You are an assistant agent in Minecraft.  I will tell you an instruction
that describes a task, and then you answer me what entity (name of blocks
or animals) I need to aim at in order to complete this task in Minecraft.
Suppose that I already have all the required inventories.  Please follow
the example below.

Task:  "chop a tree"

Thinking:  "'Chop a tree' means breaking a wood block.  So the target
entity is **wood**."

Target:  "wood"

Task:  "harvest milk"

Thinking:  "'Harvest milk' means obtaining milk from a cow.  So the
target entity is **cow**."

Target:  "cow"

Task:  "harvest wool"

Thinking:
```

The reply of ChatGPT is as follows:

```
"'Harvest wool' means obtaining wool from a sheep.  So the target entity
is sheep."

Target:  "sheep"
```

This result shows that ChatGPT understands the underlying relationship between the target item *wool* and the target animal *sheep* based on its prior knowledge about Minecraft. We also try some other instructions without telling the name of the true target entity:

```
We:

Task:  "obtain beef"

Thinking:

ChatGPT:

"'Obtain beef' means obtaining beef meat from a cow.  So the target
entity is cow."

Target:  "cow"

We:

Task:  "obtain mutton"

Thinking:

ChatGPT:

"'Obtain mutton' means obtaining mutton meat from a sheep.  So the target
entity is sheep."

Target:  "sheep"
```

Certainly, ChatGPT works on instructions containing the target entity directly:

```
We:

Task:  "harvest a flower"

Thinking:
```

```
ChatGPT:
"'Harvest a flower' means obtaining a flower from the environment.  So
the target entity is a flower."

Target:  "flower"

We:
Task:  "hunt a cow"

Thinking:

ChatGPT:
"'Hunt a cow' means hunting and killing a cow to obtain its drops, such
as beef and leather.  So the target entity is a cow."

Target:  "cow"
```

## A.2 NEGATIVE WORDS

For better segmentation, the negative word list should contain names of objects that frequently appear in Minecraft. To this end, we utilize the TF-IDF algorithm to select top-100 words from the subtitles of YouTube videos (Fan et al., 2022), excluding stop words like "we" and "is", as well as modal particles such as "yeah" and "uh". Additionally, we filter out verbs and some irrelevant nouns from the top-100 words to reduce noise. The final negative word list is shown below:

*diamond, block, village, house, iron, farm, chest, dragon, redstone, water, tree, zombie, sword, stone, door, armor, lava, fish, portal, chicken, wood, wall, glass, cave, stair, bed, torch, fire, creeper, island, food, slab, book, head, button, apple, skeleton, potion, spider, egg, pickaxe, arrow, boat, horse, hopper, box, wool, table, seed, cow, brick, trap, dog, bow, dirt, roof, leaves, sand, window, bucket, coal, hole, pig, ice, bone, stick, flower, tower, sheep, grass, sky*

Furthermore, in constructing text embeddings, we employ prompt engineering to improve zero-shot ability on classification (Radford et al., 2021). Same as MaskCLIP (Zhou et al., 2022), we utilize 85 prompt templates such as "*a photo of many* {}.". The mean of these embeddings is set to be the text embedding of the target. During segmentation, if the target object already exists in the list, it will be removed from the list in advance.

## A.3 SEGMENTATION RESULTS

We provide more examples of confidence maps, as illustrated in Figure 9. Our modified MineCLIP effectively locates these target objects.

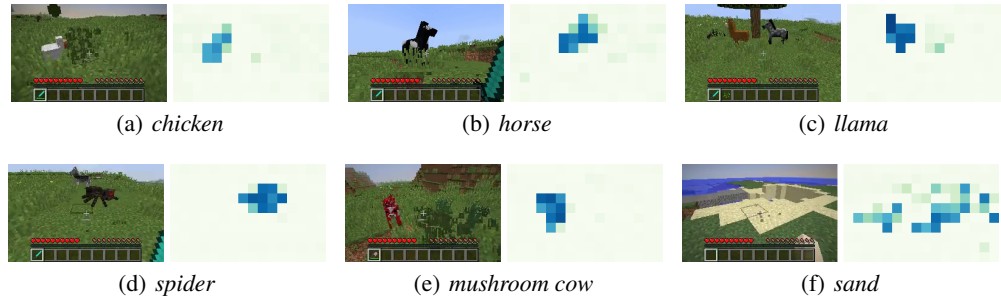

(a) *chicken*      (b) *horse*      (c) *llama*

(d) *spider*      (e) *mushroom cow*      (f) *sand*

Figure 9: Confidence map instances of targets: (a) *chicken*, (b) *horse*, (c) *llama*, (d) *spider*, (e) *mushroom cow*, and (f) *sand*.

## A.4 OFF-THE-SHELF OBJECT DETECTION MODELS

We choose two recent off-the-self object detection models, OWL-ViT (Minderer et al., 2022) and Grounding DINO (Liu et al., 2023), to evalute their effectiveness in Minecraft. As illustrated in

Figure 10 and Figure 11, both object detection models show inaccurate detection in Minecraft. Specifically, they confuse certain objects, such as cows and pigs or horses and llamas, and fail to detect some objects like sheep. This can be attributed to the significant domain gap between the stylized, blocky visuals of Minecraft and the real-world images these models were trained on. For example, pigs and sheep in Minecraft may look different from those in the real world. Therefore, we cannot directly implement these off-the-shelf object detection models in Minecraft to replace our modified MineCLIP module introduced in Section 3.1. Besides, adapting these object detection models to the Minecraft domain requires extensive labeled data with bounding boxes, which will take a lot of workforce to complete. In contrast, our modified MineCLIP inspired by MaskCLIP (Zhou et al., 2022) can realize segmentation in Minecraft without any fine-tuning.

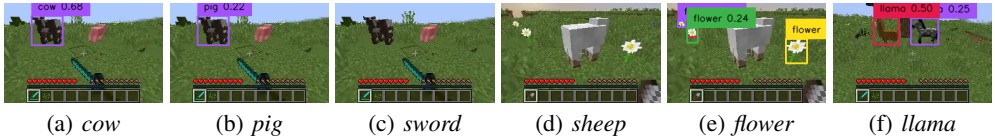

|            |          |            |            |            |            |
|------------|----------|------------|------------|------------|------------|
| (a) *cow*  | (b) *pig* | (c) *sword* | (d) *sheep* | (e) *flower* | (f) *llama* |

Figure 10: Detection results of OWL-ViT (Minderer et al., 2022).

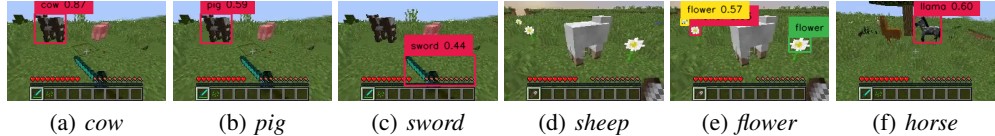

|            |          |            |            |            |            |
|------------|----------|------------|------------|------------|------------|
| (a) *cow*  | (b) *pig* | (c) *sword* | (d) *sheep* | (e) *flower* | (f) *horse* |

Figure 11: Detection results of Grounding DINO (Liu et al., 2023).

## A.5 DETAILED EVALUATION FOR GROUNDING DINO

We further evaluate the effectiveness of Grounding DINO (Liu et al., 2023) in Minecraft domain. As illustrated in Figure 12, Grounding DINO cannot discriminate between different animals. For example, for an image containing a cow and a pig, when provided with different target words, Grounding DINO consistently identifies the cow as the target. Similar situation occurs to horses and llamas. In contrast, our method exhibits clear distinctions between these animals, as demonstrated in Figure 13. In addition, the high accuracy of COPL on unseen instructions also proves our efficacy, as shown in Figure 7(c).

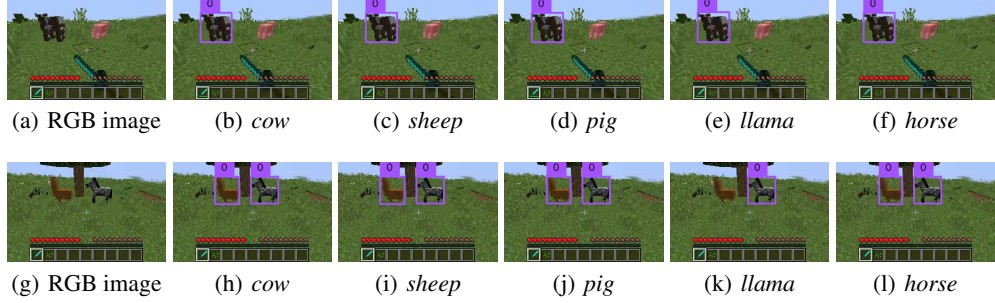

|                    |          |            |            |            |            |
|--------------------|----------|------------|------------|------------|------------|
| (a) RGB image      | (b) *cow* | (c) *sheep* | (d) *pig*  | (e) *llama* | (f) *horse* |
| (g) RGB image      | (h) *cow* | (i) *sheep* | (j) *pig*  | (k) *llama* | (l) *horse* |

Figure 12: Examples of Gounding DINO for different objects in the same image.

An approach of enhancement is to use a set of target words for object detections, similar to our constructed negative word list. Therefore, we also provide a simple word set for Grounding DINO: *cow*(0), *sheep*(1), *pig*(2), *chicken*(3), *horse*(4), *llama*(5), *spider*(6), *mushroom cow*(7), *flower*(8), *tree*(9), *sword*(10), *grass*(11). These words encompass most objects in the following evaluation images, as shown in Figure 14. The numbers in the parentheses after each word correspond to the labels of bounding boxes in the follwing figures.

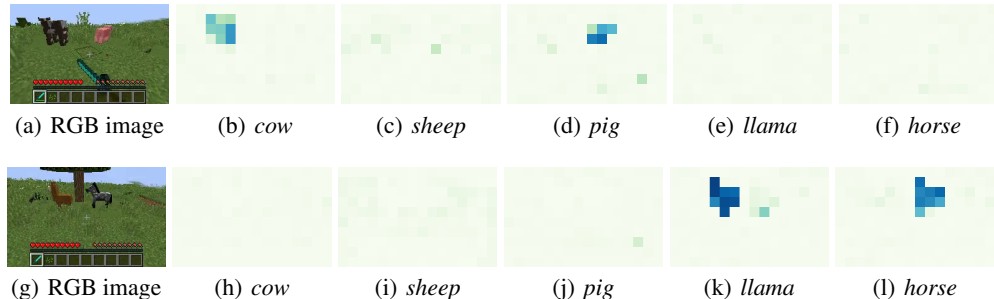

Figure 13: Examples of our method for different objects in the same image.

The results of Grounding DINO detection using this word set are demonstrated in Figure 15. Despite being provided with a word set tailored for object detection, Grounding DINO struggles to accurately identify animals. For example, it cannot identify pig, sheep, and chicken, and misidentify a mushroom cow as llama in the last image and a llama as a mushroom cow in the fourth image. To further investigate potential factors impacting detection, we conduct variations in the word set. Removing a strong background word, *grass*, from the list, as illustrated in Figure 16, does not alleviate inaccurate detection. Another variant we attempt is adding a prefix "minecraft" to each word. As reported in Figure 17, we find that such "domain prompt" does not yield improved detection results either. By these exhaustive evaluations, we show that the performance of Grounding DINO is notably limited by the domain gap between the real world and Minecraft.

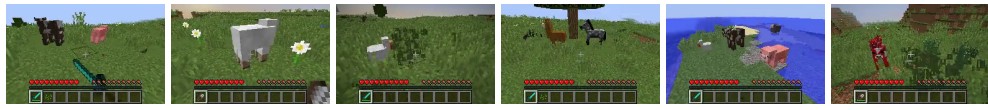

Figure 14: Orignial RGB images for evaluation.

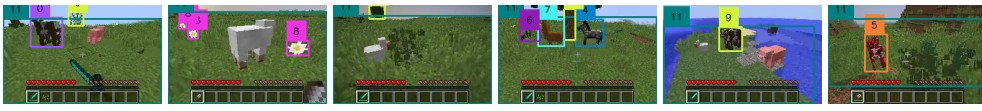

Figure 15: Gounding DINO detection results with a given word set.

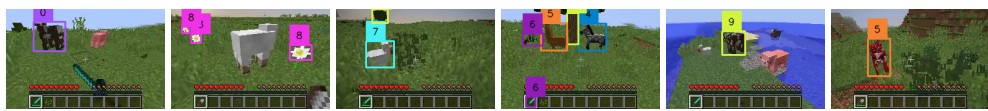

Figure 16: Gounding DINO detection results with a given word set without *grass*.

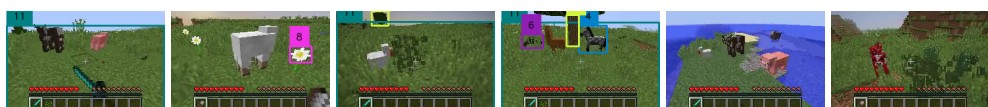

Figure 17: Gounding DINO detection results with a given word set where words are prefixed with "minecraft".

## B   POLICY LEARNING DETAILS

### B.1   OBSERVATION SPACE AND ACTION SPACE

The observation space adopted in our experiments consists of RGB, compass, GPS, voxels, and biome index. The shape and description of each modality are listed in Table 1. We simplify the orig-

inal action space of MineDojo (Fan et al., 2022) into a 2-dimensional multi-discrete action space. The first dimension contains 12 discrete actions about movement: *no_op*, *move forward*, *move backward*, *move left*, *move right*, *jump*, *sneak*, *sprint*, *camera pitch* -30, *camera pitch* +30, *camera yaw* -30, and *camera yaw* +30. The second dimension includes 3 discrete actions about interacting with items: *no_op*, *attack*, and *use*.

Table 1: Observation space adopted in our experiments.

| Modality | Shape | Description |
|---|---|---|
| RGB | (3, 160, 256) | Ego-Centric RGB frames. |
| Compass | (4,) | Sine and cosine of yaw and pitch. |
| GPS | (3,) | GPS location of the agent. |
| Voxels | (27,) | Indices of $3 \times 3 \times 3$ surrounding blocks. |
| Biome_ID | (1,) | Index of the environmental biome. |

## B.2 NETWORK ARCHITECTURE

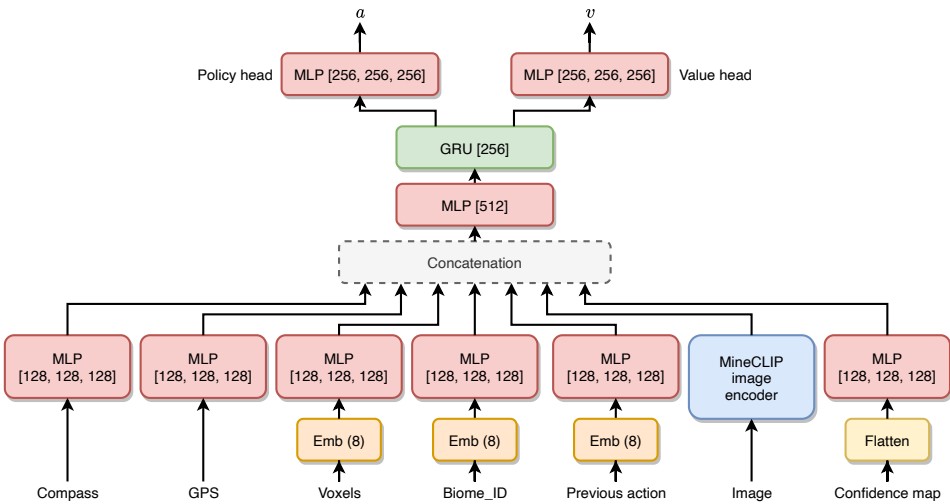

Figure 18: Network architecture of COPL agent.

The input of COPL agent includes observations from the environment listed in Table 1, the agent's action taken at last time step $a_{t-1}$, and the confidence map. As illustrated in Figure 18, all inputs except the RGB image are processed by MLPs with three hidden layers and ReLU activation. In this step, voxels, biome index, and previous action are first embedded into dense vectors. The RGB image is processed using the MineCLIP image encoder to generate an embedding. All these processed features are concatenated and processed by an MLP with one hidden layer and ReLU activation. Then a GRU layer is implemented to integrate the historical information. The policy head and the value head take as input the output of GRU and both process it using an MLP with three hidden layers and ReLU activation. The policy head generates the distribution of actions, and the value head outputs the estimated value of the current state. Some variants are as follows: (1) Single-task model: In single-task experiments, the agent *does not* take as input the confidence map; (2) EmbCLIP (Khandelwal et al., 2022): The branch of confidence map is replaced by the MineCLIP text encoder processing the target name; (3) One-Hot: The branch of confidence map is replaced by an MLP processing the one-hot vector which indicates the index of the current task. The MLP has one hidden layer with size 32 and ReLU activation.

## B.3 Environment Settings

### B.3.1 Single-Task

Our single-task experiments include four tasks: *hunt a cow*, *hunt a sheep*, *hunt a pig*, and *hunt a chicken*. The parameters we used to make environments in MineDojo are listed in Table 2. In all tasks, the agent spawns in `plains` biome holding a diamond sword. Several animals including the target spawn near the agent. The agent will receive a +100 reward after successfully killing the target animal. Each episode is limited to a maximum of 200 steps. The challenge lies in the fact that animals will flee after being attacked, thus requiring the agent to keep chasing the target and attacking. Killing a cow, sheep, or pig requires at least two attacks, while killing a chicken only requires at least one attack. Although it takes fewer attacks to kill a chicken, aiming at the small size of the chicken poses an additional challenge. For ablation experiments on Gaussian kernel, we double the initial animals and increase the animal spawn range to 10.

Table 2: Single-task settings in our experiments.

| Task | Target | Initial Animals | Range[1] | Inventory | Biome | Length[2] |
|---|---|---|---|---|---|---|
| *hunt a cow* | cow | cow, sheep, pig | 7 | diamond_sword | plains | 200 |
| *hunt a sheep* | sheep | cow, sheep, pig | 7 | diamond_sword | plains | 200 |
| *hunt a pig* | pig | cow, sheep, pig | 7 | diamond_sword | plains | 200 |
| *hunt a chicken* | chicken | cow, sheep, chicken | 7 | diamond_sword | plains | 200 |

[1] Range indicates the spawn range of initial animals.
[2] Length indicates the maximum length of one episode.

### B.3.2 More Single-Task Experiments

We conduct additional single-task experiments on three harvest tasks including *milk a cow*, *shear a sheep*, and *chop a tree*, where MineCLIP reward achieves nonzero success rates. The environment parameters for each task can be found in Table 3. As shown in Figure 19, our focal reward outperforms MineCLIP reward on *milk a cow* and *shear a sheep*. Regarding *chop a tree*, our focal reward and MineCLIP reward achieve similar performance, both with 3 out of 4 runs having learned this skill. To break a wood block, the agent needs to continuously take *attack* actions for around 6 steps. Therefore, we believe that the main challenge for RL in this task lies in exploration. It is difficult for an RL algorithm, such as PPO, with a stochastic policy to explore and exploit a behavior pattern that requires consecutive actions over 6 steps, especially given the sparse environmental reward signal. Using an off-policy RL algorithm or self-imitation may help address this problem.

Table 3: Single-task settings in additional experiments.

| Task | Target | Initial Animals | Range | Inventory | Biome | Length |
|---|---|---|---|---|---|---|
| *milk a cow* | milk_bucket | cow, sheep, pig | 10 | bucket | plains | 200 |
| *shear a sheep* | wool | cow, sheep, pig | 10 | shears | plains | 200 |
| *chop a tree* | log | cow, sheep, pig | 7 | golden_axe | forest | 200 |

### B.3.3 Multi-Task

**Hunt domain.** The hunt domain consists of four instructions: "*hunt a cow*", "*hunt a sheep*", "*hunt a pig*", and "*hunt a chicken*". At the start of each episode, one instruction is randomly selected, and an environment is built with the parameters listed in Table 4. The agent will receive a +100 reward after successfully killing the target animal specified in the instruction. If the agent mistakenly kills the animal which is the target of other instructions, no reward is given and the episode ends. This setup encourages the agent to attack the correct animal rather than indiscriminately attacking any animal. The open-vocabulary evaluation for the hunt domain also contains four instructions: "*hunt a mushroom cow*", "*hunt a spider*", "*hunt a llama*", and "*hunt a horse*". The environment parameters

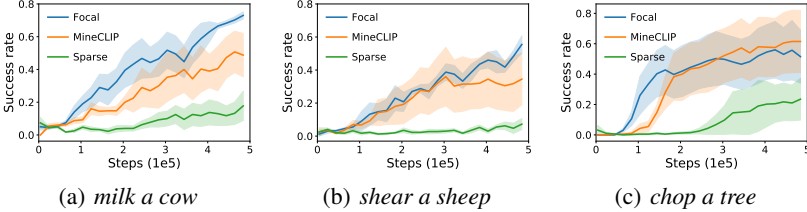

(a) *milk a cow*      (b) *shear a sheep*      (c) *chop a tree*

Figure 19: Learning curves of PPO with focal reward, MineCLIP reward, and environmental sparse reward only, on three Minecraft tasks: (a) *milk a cow*, (b) *shear a sheep*, and (c) *chop a tree*.

can be found in Table 5. We slightly increase the maximum episode length for "*hunt a spider*", "*hunt a llama*", and "*hunt a horse*", given that killing them requires more attacks as a result of their higher health compared to other animals. For each instruction, we run the test model for 100 episodes to calculate its success rate and precision (same in the harvest domain).

Table 4: Multi-task settings in the hunt domain.

| Instruction | Target | Initial Animals | Range | Inventory | Biome | Length |
|---|---|---|---|---|---|---|
| "*hunt a cow*" | cow | cow, sheep, pig, chicken | 10 | diamond_sword | plains | 500 |
| "*hunt a sheep*" | sheep | cow, sheep, pig, chicken | 10 | diamond_sword | plains | 500 |
| "*hunt a pig*" | pig | cow, sheep, pig, chicken | 10 | diamond_sword | plains | 500 |
| "*hunt a chicken*" | chicken | cow, sheep, pig, chicken | 10 | diamond_sword | plains | 500 |

Table 5: Open-vocabulary evaluation settings in the hunt domain.

| Instruction | Target | Initial Animals | Range | Inventory | Biome | Length |
|---|---|---|---|---|---|---|
| "*hunt a mushroom cow*" | mushroom cow | mushroom cow, spider, llama, horse | 10 | diamond_sword | plains | 500 |
| "*hunt a spider*" | spider | mushroom cow, spider, llama, horse | 10 | diamond_sword | plains | 800 |
| "*hunt a llama*" | llama | mushroom cow, spider, llama, horse | 10 | diamond_sword | plains | 800 |
| "*hunt a horse*" | horse | mushroom cow, spider, llama, horse | 10 | diamond_sword | plains | 800 |

**Harvest domain.** The harvest domain consists of four instructions: "*milk a cow*", "*shear a sheep*", "*harvest a flower*", and "*harvest leaves*". Same as the hunt domain, one instruction is randomly selected at the start of each episode, and an environment is generated with the parameters listed in Table 6. The agent will receive a +100 reward after successfully acquiring the target item. If the agent mistakenly acquires the target item corresponding to other instructions, no reward is given and the episode ends. Note that the target item required to finish the task may not always be the same as the target object that the agent needs to approach. For example, in the instruction "*milk a cow*", the target item is a milk_bucket, while the target object that the agent needs to approach is a cow. The environment parameters of instructions for open-vocabulary evaluation in the harvest domain can be found in Table 7. We group these instructions in terms of the inventory used to finish the task so that we can calculate the meaningful precision.

Here we introduce the behavior patterns required by the harvest domain instructions. "*Milk a cow*" and "*harvest water*" require the agent to approach the target object (cow/water), aim at it, and take *use* action. "*Harvest a flower*" and "*harvest sand*" require the agent to approach the target object (flower/sand), aim at it, take *attack* action to break it, and move closer to pick up the dropped item. "*Shear a sheep*" and "*harvest leaves*" are the same except that they require taking *use* action instead

Table 6: Multi-task settings in the harvest domain.

| Instruction | Target[1] | Initial Animals | Range | Inventory | Biome | Length |
|---|---|---|---|---|---|---|
| *"milk a cow"* | milk_bucket | cow, sheep, pig | 10 | bucket | plains | 200 |
| *"shear a sheep"* | wool | cow, sheep, pig | 10 | shears | plains | 200 |
| *"harvest a flower"* | red_flower | cow, sheep, pig | 10 | - | flower_forest | 200 |
| *"harvest leaves"* | leaves | cow, sheep, pig | 10 | shears | flower_forest | 200 |

[1] Target here represents the parameter for making a MineDojo environment, *i.e.*, the target item required to finish the task. It differs from the target object specified in the instruction.

Table 7: Open-vocabulary evaluation settings in the harvest domain.

| Instruction | Target | Initial Animals | Range | Inventory | Biome | Length |
|---|---|---|---|---|---|---|
| *"milk a cow"* | milk_bucket | cow, sheep, mushroom cow | 10 | bucket | river | 200 |
| *"harvest water"* | water_bucket | cow, sheep, mushroom cow | 10 | bucket | river | 200 |
| *"shear a sheep"* | wool | cow, sheep, mushroom cow | 10 | shears | plains | 200 |
| *"shear a mushroom cow"* | mushroom | cow, sheep, mushroom cow | 10 | shears | plains | 200 |
| *"harvest sand"* | sand | cow, sheep, mushroom cow | 10 | diamond_shovel | river | 200 |

of *attack* action. In all these tasks except "*harvest sand*", the agent needs only one *attack* or *use* action to finish the task. Therefore, individually, the harvest tasks are easier than the hunt tasks. "*Harvest sand*" requires the agent to continuously *attack* three times to break the sand block.

## B.4 HYPERPARAMETERS

**PPO Hyperparameters.** In our experiments, we use PPO as our base RL algorithm. Table 8 lists the hyperparameters for PPO across all tasks. Unlike MineAgent (Fan et al., 2022), our implementation does not include self-imitation learning and action smoothing loss. We find that vanilla PPO is able to achieve high performance in our experiments.

Table 8: Hyperparameters for PPO across all tasks.

| Hyperparameter | Value |
|---|---|
| num steps | 1000 |
| num envs | 4 |
| num minibatches | 4 |
| num epoches | 8 |
| GAE lambda | 0.95 |
| discounted gamma | 0.99 |
| entropy coef | 0.005 |
| PPO clip | 0.2 |
| learning rate | 1e-4 |
| optimizer | Adam |
| recurrent data chunk length | 10 |
| gradient clip norm | 10.0 |
| network initialization | orthogonal |
| normalize advantage | true |

**Coefficient of the intrinsic reward.** To determine the optimal scale of intrinsic reward that can effectively guide reinforcement learning while avoiding conflicts with the environmental reward, we conduct an experiment to evaluate the performance of our focal reward with different $\lambda$ values. Figure 20 illustrates the performance of our focal reward with different $\lambda$ including 0.5, 5, and 50 on *milk a cow, shear a sheep, hunt a cow* and *hunt a sheep*. Focal reward with $\lambda = 5$ outperforms $\lambda = 50$ and $\lambda = 0.5$ on all tasks. This indicates that $\lambda = 5$ is a suitable choice that demonstrates robust performance across multiple tasks and multiple domains, including both hunt and harvest domains. Therefore, we consistently set $\lambda = 5$ for all experiments in the main text.

Regarding the MineCLIP reward, we set the coefficient to 1.0, following the original setting of MineAgent. The optimal coefficient of ND reward in Tam et al. (2022) for *find* task is 0.003, and its sparse environmental reward is 1.0. However, in our experiments where the environmental reward is 100, we decided to increase the coefficient for $ND_{CLIP}$ from 0.003 to 0.3.

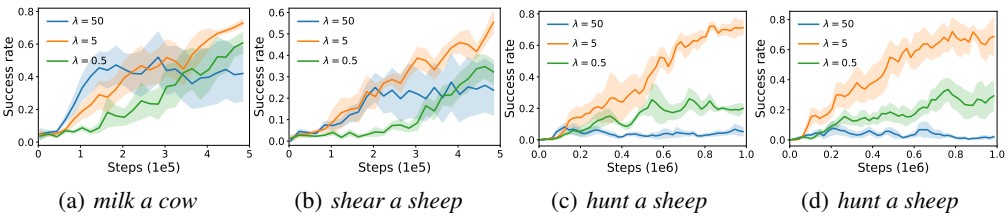

(a) *milk a cow*    (b) *shear a sheep*    (c) *hunt a sheep*    (d) *hunt a sheep*

Figure 20: Learning curves of PPO using the focal reward with different $\lambda$ on four Minecraft tasks: (a) *milk a cow*, (b) *shear a sheep*, (c) *hunt a cow*, and (d) *hunt a sheep*.

**Gaussian kernel design.** The introduction of a Gaussian kernel is to guide the agent to center a target object within its field of view. The Gaussian kernel should create a high contrast between the center and the edge, as well as between the edge and areas outside the field of view. Therefore, the variance of the Gaussian kernel would influence the performance of the focal reward. To evaluate the impact of different variances, we conduct an experiment with $\sigma = (H/5, W/5)$, $\sigma = (H/3, W/3)$, and $\sigma = (H/2, W/2)$. As illustrated in Figure 21, $\sigma = (H/3, W/3)$ outperforms the others. We suppose that a wider Gaussian kernel with $\sigma = (H/2, W/2)$ fails to provide sufficient contrast between the center and the edge. Conversely, a narrower Gaussian kernel with $\sigma = (H/5, W/5)$ cannot provide sufficient contrast between the edge and areas outside the field of view.

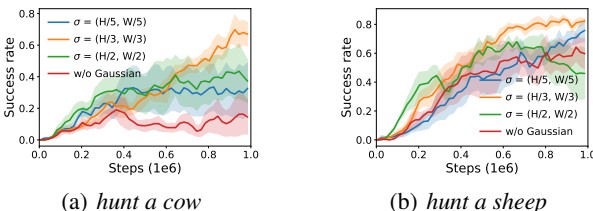

(a) *hunt a cow*    (b) *hunt a sheep*

Figure 21: Learning curves of PPO using the focal reward with different Gaussian variances on (a) *hunt a cow*, and (b) *hunt a sheep*.

### B.5 MULTI-TASK PERFORMANCE

We show the learning curves of COPL, EmbCLIP, and One-Hot on each task. As illustrated in Figures 22 and 23, the results on each task are consistent with the overall results observed in Figures 7(a) and 8(a).

Given that in our open-vocabulary evaluation for the hunt domain (Table 5 and Figures 7(b) and 7(c)), initial animals do not contain animals in training set, we conduct an additional evaluation. In this evaluation, initial animals consist of cow, sheep, pig, chicken, llama, and horse, while other settings are the same as Table 5. The results show that our model achieve 50.0(±8.1)% success rate and 88.3(±4.8)% precision for "*hunt a llama*", and 46.5(±9.8)% success rate and 90.3(±3.4)%

precision for "*hunt a horse*", indicating that the presence of training animals has no impact on the generalization performance of our model. COPL fully captures the relation between the confidence map and the task objectives, thereby unaffected by the animals it encountered during training.

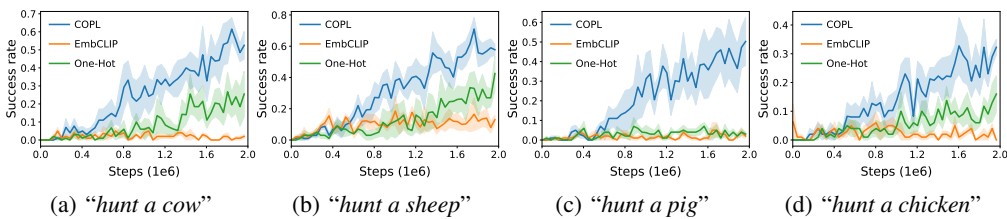

| (a) "*hunt a cow*" | (b) "*hunt a sheep*" | (c) "*hunt a pig*" | (d) "*hunt a chicken*" |

Figure 22: Learning curves of COPL, EmbCLIP, and One-Hot on four hunt instructions: (a) "*hunt a cow*", (b) "*hunt a sheep*", (c) "*hunt a pig*", and (d)"*hunt a chicken*".

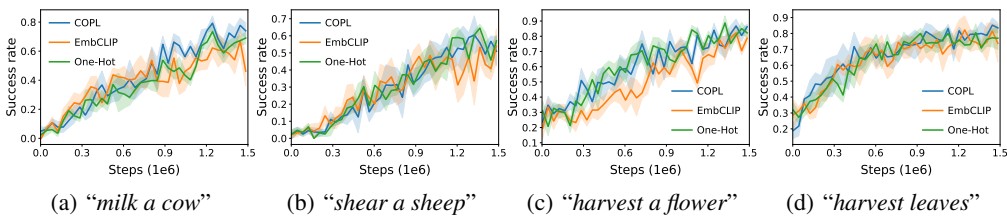

| (a) "*milk a cow*" | (b) "*shear a sheep*" | (c) "*harvest a flower*" | (d) "*harvest leaves*" |

Figure 23: Learning curves of COPL, EmbCLIP, and One-Hot on four harvest instructions: (a) "*milk a cow*", (b) "*shear a sheep*", (c) "*harvest a flower*", and (d)"*harvest leaves*".

## B.6   BASELINES IMPLEMENTATION

**MineCLIP.** We adopt the provided prompt templates in MineDojo to design task prompts for MineCLIP reward computation. For hunt tasks, we use the prompt "*hunt a {animal} on plains with a diamond sword*". For harvest tasks in Appendix B.3.2, we use the prompts "*obtain milk from a cow in plains with an empty bucket*", "*shear a sheep in plains with shears*", and "*chop trees to obtain log with a golden axe*", respectively.

**Cai et al. (2023).** We use the released plains model[2] for evaluation.

**STEVE-1.** We use the released model[3] for evaluation. However, STEVE-1 (Lifshitz et al., 2023) is designed for another simulator, MineRL (Guss et al., 2019), with a different action space from Mine-Dojo. We build a wrapper to map STEVE-1's actions into the action space of MineDojo. As noted in the STEVE-1 paper, prompt engineering significantly impacts its performance. Therefore, we attempt three templates for the hunt domain tasks including "*kill a {animal}*", "*hunt a {animal}*", and "*combat a {animal}*". As shown in Figure 24, "*kill a {animal}*" achieves the highest performance and STEVE-1 cannot understand the original instruction

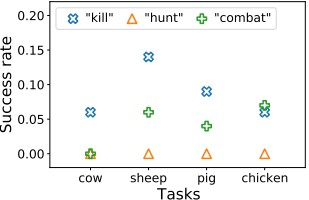

Figure 24: Success rates of STEVE-1 with different prompts.

"*hunt a {animal}*" at all. Consequently, we use "*kill a {animal}*" as prompts given to STEVE-1 for the experiments in the main text. For tasks in the harvest domain, we use prompts "*milk a cow*", "*shear a sheep*", "*break a flower*", "*break leaves*", "*collect water*", "*shear a mushroom*", and "*collect sand*", respectively. The verbs *break* and *collect* are selected by referring to the prompts

---

[2] https://github.com/CraftJarvis/MC-Controller
[3] https://github.com/Shalev-Lifshitz/STEVE-1

provided in the STEVE-1 paper. "*Milk a cow*", "*shear a sheep*", and "*shear a mushroom cow*" follow original instructions, as we find that "*collect {milk/wool/mushroom}*" does not work.

## C    SUPPLEMENTARY EXPERIMENTS

### C.1    GROUNDED SAM EVALUATION

In order to conduct a side-by-side comparison between Grounded SAM (Liu et al., 2023; Kirillov et al., 2023) and our method, we Google searched "minecraft [object name] screenshot" in image tab, and selected the first two images that includes objects and has them fully in field of view. The evaluation objects includes *pig*, *cow*, *sheep*, *mushroom cow*, *tree*, *flower*, *horse*, and *torch*. We follow the setting in the official demo[4] to evaluate the effectiveness of Grounded SAM on detecting these objects in Minecraft. Additionally, we include a negative class "grass". For reference, we also provide the 2D confidence maps generated by our method.

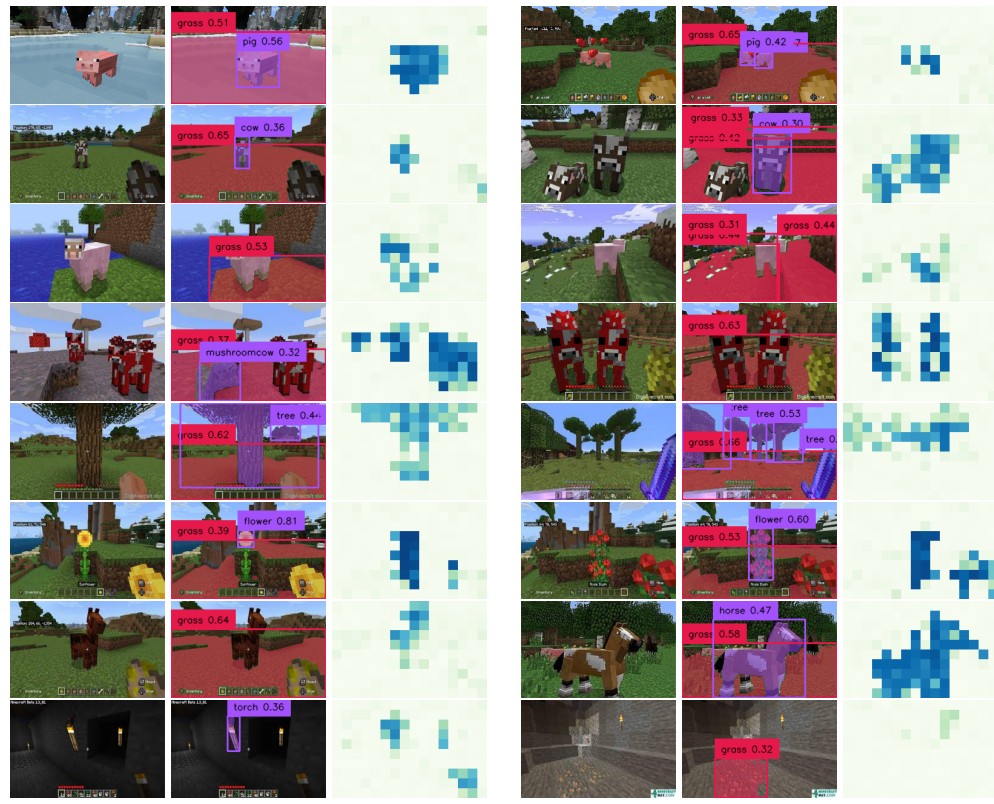

Figure 25: Comparison between Grounded SAM and our method on eight objects.

The detection results of two methods are illustrated in Figure 25. For a more detailed evaluation, we quantify the number of objects present in each image, the number detected by Grounded SAM, and the number detected by our method. These quantitative results are summarized in Table 9. Across all images, there are 28 target objects in fact. Grounded SAM is able to successfully identify 15 objects, which translates to a detection rate of 53.6%. In contrast, our method demonstrates a significantly

---

[4]https://github.com/IDEA-Research/Grounded-Segment-Anything/blob/main/grounded_sam_colab_demo.ipynb

higher efficacy, successfully detecting 26 of the 28 objects, achieving a detection rate of 92.9%. There are two failures in our method. One is the sunflower in the bottom-right corner of the first flower image, and the other is the torch in the second torch image. In both cases, our method actually generates some activation in the target patches, but it does not cover the entire object (flower) or is relatively weak (torch). We regard them as failures for a more strict result.

Table 9: Detection result statistics of Grounded SAM and our method on eight objects.

| objects | # of objects | # of Grounded SAM | # of ours |
|---|---|---|---|
| *pig* | 3 | 3 | 3 |
| *cow* | 3 | 2 | 3 |
| *sheep* | 2 | 0 | 2 |
| *mushroom cow* | 4 | 0 | 4 |
| *tree* | 6 | 6 | 6 |
| *flower*[†] | 4 | 2 | 3 |
| *horse* | 2 | 1 | 2 |
| *torch* | 4 | 1 | 3 |
| **total** | **28** | **15** | **26** |

[†] We also count the two flowers held in players' hands.

We also conduct a single-task experiment on *shear a sheep* to compare the effectiveness of the reward calculated by our method and by Grounded SAM quantitatively. In detail, we provide Grounded SAM with the observation image and the target object name "sheep", along with a negative class "grass", and obtain a segmentation map of the target "sheep". This map is then used in place of $m_t^c$ in Equation (1) to calculate the reward. As illustrated in Figure 26, the performance of Grounded SAM is only similar to that of sparse reward, implying that the segmentation map generated by Grounded SAM is noisy and could not provide accurate location information of about target object "sheep".

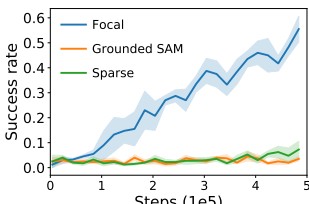

Figure 26: Learning curves of PPO with focal reward, Grounded SAM reward and environmental sparse reward only, on task *shear a sheep*.

## C.2 ANIMAL ZOO EXPERIMENT

We conduct a multi-task experiment following the Animal Zoo setting in the original MineCLIP paper (Fan et al., 2022). This experiment aims to compare the effectiveness of our focal reward and the MineCLIP reward in multi-task RL. The detailed settings are listed in Table 10. Consistent with the original paper, we implement the policy network taking as input the whole task prompt, rather than a confidence map or only the target name.

Table 10: Multi-task settings in Animal Zoo.

| Prompt | Initial Animals | Range | Inventory | Biome | Length |
|---|---|---|---|---|---|
| "*hunt a cow*" | cow, sheep, pig | 10 | diamond_sword | plains | 500 |
| "*hunt a sheep*" | cow, sheep, pig | 10 | diamond_sword | plains | 500 |
| "*milk a cow*" | cow, sheep, pig | 10 | bucket | plains | 500 |
| "*shear a sheep*" | cow, sheep, pig | 10 | shears | plains | 500 |

The results are shown in Figure 27. The MineCLIP reward guides the agent to complete two easier tasks, *milk a cow* and *shear a sheep*, but fails in two more difficult hunting tasks. In contrast, our focal reward enables the agent to achieve significant success rates in all tasks. On the one hand, our focal reward improves the learning speed on two easier tasks. On the other hand, our focal reward notably outperforms the MineCLIP reward on two harder tasks where the MineCLIP reward fails. These results, along with those from single-task experiments in Figure 5, demonstrate the superiority of our focal reward over MineCLIP reward. We also conduct additional tests to evaluate the generalization ability of the multi-task policy learned with our focal reward. The policy achieves a 0(±0)% success rate for the prompt "*hunt a pig*" and 2.3(±1.9)% success rate for "*hunt a chicken*", suggesting that four training language prompts or instructions are insufficient for learning an open-vocabulary policy.

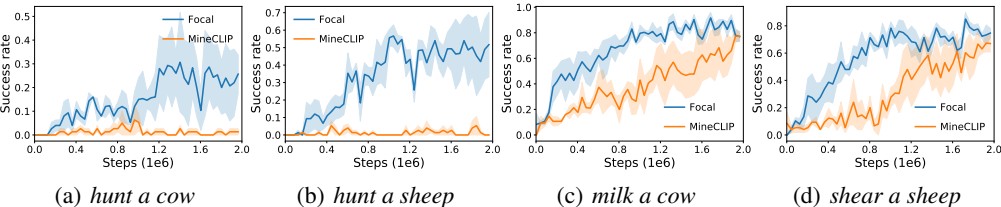

| (a) *hunt a cow* | (b) *hunt a sheep* | (c) *milk a cow* | (d) *shear a sheep* |

Figure 27: Learning curves of multi-task PPO with focal reward and MineCLIP reward on four Animal Zoo tasks.

