# OpenReview forum: "CLIP-Guided Reinforcement Learning for Open-Vocabulary Tasks"
_ICLR.cc/2024/Conference — Submitted to ICLR 2024_

### Official Review · Reviewer_VdeC · 2023-10-30

**Soundness:** 3 good
**Presentation:** 3 good
**Contribution:** 3 good
**Rating:** 6
**Confidence:** 4

**Summary:**

The paper proposes a new reward function and task specification strategy for RL on Minecraft. First, ChatGPT is used to extract a target object from a given language instruction. Then, a modified version of MineCLIP is used to convert the target object and current observation image into a segmentation map that highlights the location of the target object in the image. The authors propose a shaped reward function that incentivizes both increasing the area of the target object in the segmentation map and centering the target object in the frame. Instead of conditioning the policy on the language instruction, the policy is conditioned on the segmentation map. PPO is used to optimize the policy against the proposed shaped reward (plus the original sparse reward). Experiments show that this method outperforms prior methods (STEVE-1, Cai et al) and ablations on language-conditioned Minecraft tasks. Additionally, the experiments show that the learned policy can generalize zero-shot to new instructions.

**Strengths:**

The paper addresses reinforcement learning of open-world, open-vocabulary instruction following which is a problem of significant interest to the community. Building on prior work, the authors use Minecraft as a test-bed for their method. Minecraft is becoming a standard benchmark for these types of methods and so this choice will allow for easier comparison to prior work. The proposed method is a novel modification of existing work (MineCLIP). The motivation and explanation of the method is clear. The experiments ablate the different components of the method and compare to prior work.

**Weaknesses:**

My main concern is that this method makes more assumptions than the prior work it is compared to. Specifically, this method assumes the task involves navigating to a target object that is specified in the instruction. An example of an instruction where this method would not work as well is "build a tower" (since there is no target object to move toward). Notably, MineCLIP, STEVE-1, and Cai et al do not make this assumption.

Additionally, the comparison to imitation learning methods like STEVE-1 and Cai et al should be justified since the proposed method does online RL. Do the imitation learning methods see more or less relevant data for the evaluations tasks? Imitation learning and online RL are different classes of methods so some explanation is needed here.

Smaller comments:
- Throughout the paper it seems that "open-vocabulary" is used to mean "unseen instructions" For instance, the "open-vocabulary" section in the experiments describes testing generalization to unseen instructions. While a method must be open-vocabulary to accept unseen instructions, it would be more clear to specifically state that the capability these experiments test is zero-shot generalization to unseen instructions.
- In Figures 7 and 8 it would be good to indicate in the plots which tasks involve unseen instructions and which just involve an unseen biome (since it seems both are tested).
- In Figure 8, it seems like the success rate plots (b and c) could be combined (with some indication of which tasks involve unseen instructions/biomes) like in Figure 7.

**Questions:**

- Why wasn't EmbCLIP evaluated on unseen instructions in the hunt domain?
- Why is "shear a sheep" considered an unseen instruction when it's part of the instructions seen during training? ("open vocabulary generalization" section, second paragraph).

---

> ### Author Response · Authors · 2023-11-15
>
> Thank you for your efforts and valuable comments. We would like to address the weaknesses and questions you raised in the review.
>
> > Why wasn't EmbCLIP evaluated on unseen instructions in the hunt domain?
>
> We do not evaluate EmbCLIP on unseen instructions in the hunt domain due to its limited performance on the training instructions. Since the model does not perform well on tasks it is trained for, we determine that it would not provide meaningful insights to test it on unseen instructions.
>
> > Why is "shear a sheep" considered an unseen instruction when it's part of the instructions seen during training?
>
> We would like to clarify that 'shear a sheep' is not considered as an unseen instruction here. As shown in Figure 8, we do not highlight 'wool' in green. **The inclusion of 'shear a sheep' in the generalization test is used to calculate the precision**. As defined in the same paragraph, precision is calculated as the number of times correctly harvesting the specified target divided by the number of times harvesting any target declared in the group’s instructions. Therefore, 'shear a sheep' is grouped with 'shear a mushroom cow' since both tasks require the tool 'shears'. This grouping allows us to assess the precision of the model in correctly identifying the unseen target object 'mushroom cow', ensuring that the agent does not simply shear any object indiscriminately. Similarly, another training instruction 'milk a cow' is also included in a group along with an unseen instruction 'harvest water' for the same reason.
>
> > Comparison to imitation learning methods.
>
> In our study, the comparison between COPL and the imitation learning methods, STEVE-1 and Cai et al., is due to the lack of alternative Minecraft foundation models with goal- or instruction-conditioned policies trained via reinforcement learning. We select STEVE-1 and Cai et al. for their capabilities in mastering multiple basic skills in Minecraft, similar to the objectives of our open-vocabulary evaluation.
>
> We acknowledge that comparing an online RL method with imitation learning methods is not a perfect match since the training data is quite different. Therefore, our intention is not to determine which method performs best in learning basic skills in Minecraft. Instead, we just use STEVE-1 and Cai et al. as a reference to understand how COPL performs in this specific domain.
>
> > Smaller comments.
>
> Thank you for your detailed suggestions. We will make improvements based on them in the revision.
>
> ----
>
> If you have any additional questions or comments, please do not hesitate to let us know. We are more than happy to provide any further clarifications or information that may be helpful. If you find our responses satisfactory, we would be grateful if you could consider raising the score.

---

> > ### Comment · Reviewer_VdeC · 2023-11-20
> >
> > Thank you for answering my questions.
> >
> > However, I remain concerned that this method makes more assumptions about the task than the prior work it is compared to. As the other reviewers have pointed out, this proposed reward is only valid for tasks that involve moving towards a target object while the prior methods work for more general Minecraft tasks (MineClip, STEVE-1, Cai et. al). This assumption is clear in the choice of tasks for the experiments (only hunting and harvesting tasks). Additionally, the proposed method only outperforms much simpler baselines (one-hot and CLIP embedding conditioning) in the hunting domain.
> >
> > While it's fine to propose a method with more assumptions and more limited applicability, the paper should make those assumptions clear. The paper currently claims that the proposed method solves open-vocabulary tasks, implying the method is applicable to all tasks that can be specified with language.

---

> ### Author Response · Authors · 2023-11-21
>
> Dear reviewer, we are happy to receive your response and appreciate the opportunity to further discuss and address your concern.
>
> > While it's fine to propose a method with more assumptions and more limited applicability, the paper should make those assumptions clear. The paper currently claims that the proposed method solves open-vocabulary tasks, implying the method is applicable to all tasks that can be specified with language.
>
> In the original version of our paper, we claimed "By open-vocabulary task, we mean that the agent is instructed to interact with diverse objects beyond the training scope." at the beginning of Section 2, by which we limit the scope of tasks that our method is designed to solve. We acknowledge, however, that this description may not be precise enough. Therefore, based on the comments from reviewers, we append an additional statement to provide a more restrictive claim on our assumption and application: **"More specifically, we focus on object-centric tasks and the open-vocabulary ability over target objects."**, in the latest revision. Thank you for bringing this to our attention, and we hope this clarification aligns with the paper's objectives more accurately.
>
> > Additionally, the proposed method only outperforms much simpler baselines (one-hot and CLIP embedding conditioning) in the hunting domain.
>
> Firstly, we would like to highlight that our proposed method outperforms baselines not only in the hunt domain but also in the harvest domain. While the learning curves of the three models on **training instructions** appear comparable, as shown in Figure 8(a), the gap between our method and the baseline is significant on **unseen instructions** during test, as illustrated in Figure 8(c). This result demonstrates our method's superior ability to generalize to unseen instructions than EmbCLIP, suggesting that our proposed approach of taking 2D confidence map as a policy conditioning representation offers better open-vocabulary ability over target objects compared to relying on language descriptions.
>
> Secondly, we think that using a one-hot vector as the task indicator (One-Hot) and conditioning policy on language (EmbCLIP) are not simple, but are essential instead. COPL, EmbCLIP, and One-Hot share the same network architecture and training procedures, thus enabling a clear and direct comparison of generalization ability between policy conditioning representations from different modalities (vision, language, and vector), excluding the influence of other factors.
>
> > ... this proposed reward is only valid for tasks that involve moving towards a target object while the prior methods work for more general Minecraft tasks (MineClip, STEVE-1, Cai et. al).
>
> We acknowledge the limitation of COPL in the aspect of generality, as we newly added description in Section 2. At last, we would like to offer a comprehensive perspective that combines both theoretical and empirical considerations here.
>
> **MineCLIP.** Although theoretically, MineCLIP can align video clips with a wide range of language descriptions, besides object-centric tasks, previous work [1,2] and our experiments demonstrate its practical limitations. We find that in practice, MineCLIP still tends to align entities in video clips with those in language descriptions like the original CLIP, instead of utilizing temporal information (otherwise the MineCLIP reward would effectively guide the agent to approach targets in object-centric tasks).
>
> **Imitation learning.** STEVE-1 struggles with accurately differentiating targets, as evidenced by its low performance in the hunt domain. Cai et al. (2023) show limited generalization ability over target objects, as illustrated in Figure 8(b), due to its condition on language input. This aligns with our observations on EmbCLIP. Theoretically, we can attribute these deficiencies to the limitation of the imitation learning dataset, instead of the methods themselves. However, from another perspective, reinforcement learning shows advantages here of being independent of the quality and distribution of the demonstration dataset.
>
> In summary, no method can perfectly handle all tasks and perform zero-shot generalization. Our method could be regarded as a practical compromise that **sacrifices some of MineCLIP's theoretical capabilities of generality but enhances the practical performance for object-centric tasks**, which compose a large portion of tasks in Minecraft.
>
> [1] Cai et al., 2023, Open-World Multi-Task Control Through Goal-Aware Representation Learning and Adaptive Horizon Prediction.
>
> [2] Yuan et al., 2023, Plan4MC: Skill Reinforcement Learning and Planning for Open-World Minecraft Tasks.

---

### Official Review · Reviewer_NgQ4 · 2023-10-31

**Soundness:** 2 fair
**Presentation:** 2 fair
**Contribution:** 2 fair
**Rating:** 5
**Confidence:** 4

**Summary:**

The paper introduces a method for improving CLIP-guided rewards for Reinforcement Learning for completing open-vocabulary tasks within the game of Minecraft.

The approach builds on top of MineCLIP, a VLM trained on internet-scale Minecraft videos, used as an auxiliary reward model for training PPO to solve Minecraft tasks expressed in the form of natural language. Instead of using the original MineCLIP image embedding and computing its individual cosine distance with the MineCLIP language encoding of the task to obtain a reward, the authors propose leveraging recent techniques for open-vocabulary segmentation with CLIP to produce a “2d confidence map” for the open-vocabulary task target over the image visual field. This 2d confidence map is computed by obtaining CLIP embeddings for “patches” of the image, and obtaining the normalized cosine distance of each patch with the embedding for the target text (with a subsequent “denoising” step making use of several negative prompts). This 2d map is used as an additional input to the policy MLP. Moreover, to then obtain a reward, one must integrate over this 2d confidence map, weighting the entries with a gaussian kernel (obtaining a “focal” reward). This reward is then multiplied by a constant and summed to the vanilla environment reward.

The authors conduct several experiments on tasks belonging to the “hunting” domain, comparing their baseline (COPL) with other techniques for auxiliary rewards, such as the original MineCLIP. They also test the generalization of their method on tasks belonging to the “hunting” and “harvesting” domains, generalizing the target of the task to unseen objects in an open-domain fashion. Over these experiments, COPL reliably performs better than alternatives.

**Strengths:**

The proposed method addresses a specific shortcoming of MineCLIP, a previous approach for auxiliary semantic rewards for Minecraft tasks. Essentially, the problem with MineCLIP is that it serves as a very noisy and not well shaped reward signal for language based tasks. A well-shaped RL reward for several Minecraft tasks should involve distance to a target object, and COPL fixes this problem with its 2d confidence map technique.

The experiments seem to show a clear improvement over baselines for the chosen hunting tasks, showing that COPL indeed works as a better-shaped CLIP reward for such tasks.

**Weaknesses:**

Overall, the main problem with the approach is that it does not seem to be very “general”. This would not be a problem per se (not all ICLR papers should aim at “general” solutions to problems), if not for the fact that the work builds directly on top of MineCLIP, which was aimed at producing a multi-task “general” agent for open-ended Minecraft tasks.

To be more specific: the original MineDojo paper involved attempting to solve all kinds of Minecraft tasks based on their language descriptions (“milk a cow”, “hunt a sheep”, “combat a zombie”, “find a nether portal”, “dig a hole”, “lay a carpet”). For this reason, they proposed a “general” method making use of a MineCLIP encoder, encoding image sequences and language commands, which is not limited to a specific “task domain”. The MineCLIP encoder can in principle encode image sequences for every Minecraft task, be it hunting, combat, pure world exploration, or tasks that do not involve focusing on a game “entity”, such as simply digging a hole. (Whether it achieved satisfactory results is another matter)

In this paper, the MineCLIP encoder is instead taken as a building block, to then do image segmentation based on individual “entity” labels such as “cow”, “pig” and “sheep”. This means that essentially, in order to improve performance on some specific task domains such as hunting, the method’s generality was reduced to be only suitable for tasks involving focusing on and getting closer to specific game entities (it is no longer possible to conceivably use this technique to learn the task “dig a hole”, or “lay a carpet”). Essentially, the “COPL” technique consists of a method for turning a suitable open-vocabulary image segmentation model into a 2d confidence map that helps both as a better task-conditioned input for the policy MLP, and as a more well-shaped reward model, biased towards looking at entities and getting close to them.

In the experiment section, most results that paint the COPL method in a clearly positive light belong to the “hunt” domain. Essentially, what needs to be learnt within this domain is to identify an entity in the world based on a word, keep it within the center of the screen, attack it and pursue it while it flees. It is apparent why the specific biases of the focal COPL reward function would help in this case, to the point it could be considered as “overfitted” to tasks similar to this. For the “harvesting” domain (which still involves in practice finding a specific “entity” in the world, getting close to it, and collecting it), the benefits of the technique already appear smaller or not present. No other open-ended tasks have been tried, and it’s doubtful that the COPL technique would even be applicable for them (how to do so for “dig a hole”?).

What I’m getting at is, if the specific domains and settings for this method to be useful have to be so restricted, what stops us from directly using traditional reward shaping in the state space of the Minecraft world (not general purpose, but strong)? In any case, we no longer support free-form text prompts and are overfitted to hunting tasks. I would appreciate further elaboration on this point.

**Questions:**

I have the following questions:
* Could you elaborate on the experiment design for the experiment in Figure 8? Why are learning curves so similar for all methods in panel (a), but not in the panels (c) and (d), where MineCLIP seems to perform worse than COPL? Why is there no “one hot” in panels (b), (c) and (d)?
* From a cursory look, it seems that the MineCLIP baseline agent for tasks such as “hunt a cow” seems to severely underperform relative to the one from the original MineCLIP paper. Can you comment on this?

---

> ### Author Response · Authors · 2023-11-15
>
> Thank you for your efforts and valuable comments. We appreciate such a detailed and insightful discussion on our method. We would like to address the weaknesses and questions you raised in the review.
>
> > Why are learning curves so similar for all methods in panel (a), but not in the panels (c) and (d), where MineCLIP seems to perform worse than COPL?
>
> There might be a typo in the question, as 'MineCLIP' is not mentioned in Figure 8; we assume 'EmbCLIP' is intended.
>
> In Figure 8(a), the learning curves show the success rates of three models on **training instructions** throughout the training process. The similarity in these curves indicates that all methods have comparable performance on the training tasks. The results demonstrated in Figure 8(c) and (d) include training instructions and **unseen instructions** (highlighted in green). The gap between COPL and EmbCLIP on training instructions ('milk a cow' and 'shear a sheep') is narrow and approximately aligns with the gap observed at the end of the training in Figure 8(a). However, the gap widens on unseen instructions ('harvest water', 'shear a mushroom cow', and 'harvest sand'), suggesting COPL's superior generalization ability.
>
> Specifically, for EmbCLIP, while the text encoder can produce a language embedding for the novel target object, this embedding is essentially out-of-distribution for the policy network. This is because the language embedding space is huge but the policy network is only trained on a limited set, *i.e.*, embeddings of four training target objects. In contrast, COPL's policy network takes a 2D confidence map as input. After our modified MineCLIP converts the unseen target name into a confidence map, the policy network can process it without a generalization gap.
>
> We would like to highlight this novel use of a 2D confidence map as a policy conditioning representation is another contribution of our work, besides the intrinsic reward. Multi-task RL with the original MineCLIP model cannot acquire such a unified representation but only takes as input the language instruction, as implemented in the original MineCLIP paper.
>
> The elaborated experiment settings are available in Appendix B.3.3.
>
> > Why is there no 'one hot' in panels (b), (c) and (d)?
>
> No One-Hot in Figure 8(b): Figure 8(b) focuses on training instructions. Since COPL, EmbCLIP, and One-Hot exhibit similar performance on these training instructions, we omit EmbCLIP and One-Hot here for brevity. We provide success rates of these methods on each training instruction in Figure 17.
>
> No One-Hot in Figure 8(c) and 8(d): Figure 8(c) and 8(d) are intended to present generalization performance on unseen instructions. However, One-Hot is only a multi-task baseline and cannot perform meaningful generalization test. In detail, it uses a one-hot vector as the task indicator, which is set to the same length as the number of training tasks. This design means that One-Hot lacks the flexibility to handle unseen instruction, as there is no appropriate task indicator.
>
> > From a cursory look, it seems that the MineCLIP baseline agent for tasks such as “hunt a cow” seems to severely underperform relative to the one from the original MineCLIP paper. Can you comment on this?
>
> We appreciate your observation of this noticeable phenomenon that we did not mention in the paper but is worthy of analysis.
>
> In the original MineCLIP paper, four tasks, 'milk a cow', 'hunt a cow', 'shear a sheep', and 'hunt a sheep', are trained together under a multi-task reinforcement learning framework. Among these four tasks, 'milk a cow' and 'shear a sheep' are relatively easier to learn than the others. Therefore, the MineCLIP reward likely guides the agent to successfully learn these easier tasks first, and then a behavior bias, moving towards a cow or sheep, is introduced into the policy network, facilitating the learning of more challenging hunt tasks. However, in our hunt task learning experiments, there are no such easy tasks to introduce behavior bias, resulting in the underperformance of MineCLIP reward. Other minor reasons include the fact that we do not use action smoothing and self-imitation learning and we employ a different PPO codebase.
>
> Another noticeable point is that, as shown in its Table 1, multi-task RL in the original MineCLIP paper seems to overfit to cow-related tasks, possibly due to the use of language instructions as input. This challenge of correctly grounding language into visual targets is also observed in our experiment (EmbCLIP in Figure 7(a)). This demonstrates the advantage of our method using a unified 2D confidence map as the policy condition.
>
> Additionally, Figure 13 demonstrates that our implementation of the MineCLIP reward successfully guides the agent to master harvest tasks, including 'milk a cow' and 'shear a sheep', confirming the correctness of our implementation.

---

> ### Author Response · Authors · 2023-11-15
>
> > For the “harvesting” domain, the benefits of the technique already appear smaller or not present.
>
> A detailed explanation of the open-vocabulary evaluation in the harvest domain is in the answer to the first question.
>
> > ... what stops us from directly using traditional reward shaping in the state space of the Minecraft world (not general purpose, but strong)?
>
> If the goal is only to learn tasks in Minecraft, we agree that using traditional reward shaping in the state space of the Minecraft world, such as the 'manual reward' designed in the original MineCLIP paper, is a direct approach. But from our perspective, we wish our proposed methodology applicable in diverse environments, rather than only in Minecraft, for the problem defined in Section 2. As you succinctly summarized, "Essentially, the 'COPL' technique consists of a method for ... getting close to them", the whole framework is not tailored for Minecraft and can also be applied to other similar first-person view and object-centric environments such as robotic navigation. We use Minecraft as a convenient and suitable testbed to validate our method. Therefore, we avoid the use of environment-specific information to guide reinforcement learning.
>
> > Comparison with MineCLIP in terms of generality.
>
> As mentioned above, our method could be viewed as a framework applicable to diverse environments, not limited to the specific use case of Minecraft. In Minecraft, we use MineCLIP to realize the first component of our method, "turning a suitable open-vocabulary image segmentation model into a 2d confidence map". We do not intend to build a new fundamental model or method that theoretically surpasses MineCLIP in terms of multimodal ability and generality in Minecraft. Instead, it just serves as a key module specific to Minecraft within our framework to solve the problem we defined in Section 2. In other first-person view and object-centric environments, our framework can adapt by replacing the modified MineCLIP with general models like Grounding DINO + SAM or domain-specific models.
>
> ----
>
> If you have any additional questions or comments, please do not hesitate to let us know. We are more than happy to provide any further clarifications or information that may be helpful. If you find our responses satisfactory, we would be grateful if you could raise the score.

---

> > ### Comment · Reviewer_NgQ4 · 2023-11-21
> > **Response to the Authors**
> >
> > Dear Authors,
> >
> > I thank you for your detailed reply. I still have concerns, namely, citing your reply:
> >
> > > Specifically, for EmbCLIP, while the text encoder can produce a language embedding for the novel target object, this embedding is essentially out-of-distribution for the policy network. This is because the language embedding space is huge but the policy network is only trained on a limited set, i.e., embeddings of four training target objects. In contrast, COPL's policy network takes a 2D confidence map as input. After our modified MineCLIP converts the unseen target name into a confidence map, the policy network can process it without a generalization gap.
> >
> > This is clear, however this implies that all information contained in the language embedding is essentially lost, except for a 2d scalar map, which is only useful insofar as the language task description mentions an object that the agent needs to get close to. If we expect this embedding to be useful to condition policies for any other task (such as "place a carpet"), we have lost all information. This is what I meant when I said that the proposed method reduces the generality of MineCLIP.
> >
> > > In the original MineCLIP paper, four tasks, 'milk a cow', 'hunt a cow', 'shear a sheep', and 'hunt a sheep', are trained together under a multi-task reinforcement learning framework. Among these four tasks, 'milk a cow' and 'shear a sheep' are relatively easier to learn than the others. Therefore, the MineCLIP reward likely guides the agent to successfully learn these easier tasks first, and then a behavior bias, moving towards a cow or sheep, is introduced into the policy network, facilitating the learning of more challenging hunt tasks. However, in our hunt task learning experiments, there are no such easy tasks to introduce behavior bias, resulting in the underperformance of MineCLIP reward. Other minor reasons include the fact that we do not use action smoothing and self-imitation learning and we employ a different PPO codebase.
> >
> > Your reply here leads me to believe that your approach should fit in seamlessly in the multitask RL setting from the original paper, and thus better comparisons would have been obtained if starting from such a multitask baseline, adding your focal reward technique on top.
> >
> > I read and understood your further points on how you were not planning on introducing "a new fundamental model or method that theoretically surpasses MineCLIP in terms of multimodal ability and generality in Minecraft". Still, even understanding this, the paper's contribution seem underwhelming, as they boil down to the use of a focal reward function based on a 2d object detection map.

---

> > > ### Author Response · Authors · 2023-11-23
> > >
> > > Dear reviewer,
> > >
> > > Thank you for your response.
> > >
> > > > If we expect this embedding to be useful to condition policies for any other task (such as "place a carpet"), we have lost all information. This is what I meant when I said that the proposed method reduces the generality of MineCLIP.
> > >
> > > We append an additional statement in Section 2 "Problem Statement" to provide a more restrictive claim on our assumption and application: "More specifically, we focus on object-centric tasks and the open-vocabulary ability over target objects.", in the revision. Given the limited practical performance of MineCLIP reward in our experiments, our method could be regarded as a practical compromise that sacrifices some of MineCLIP's theoretical capabilities of generality but enhances the practical performance for object-centric tasks, which compose a large portion of tasks in Minecraft.
> > >
> > > > ... thus better comparisons would have been obtained if starting from such a multitask baseline, adding your focal reward technique on top.
> > >
> > > Following your suggestion, we conduct a multi-task experiment based on the Animal Zoo setting introduced in the original MineCLIP paper, including 'milk a cow', 'hunt a cow', 'shear a sheep', and 'hunt a sheep'. As shown in Figure 27, our focal reward outperforms the MineCLIP reward across all tasks. We believe this experiment could serve as a better and more convincing comparison.
> > >
> > > We note that the results of the MineCLIP reward are not the same as those in the original paper. Results in the original paper exhibit high success rates in 'hunt a cow' and 'milk a cow' while ours show high success rates in 'milk a cow' and 'shear a sheep'. This disparity suggests that the influence of the fact that we do not use action smoothing, self-imitation learning, and multi-stage training might not be that 'minor' as we mentioned in previous response.

---

### Official Review · Reviewer_9Xm8 · 2023-10-31

**Soundness:** 4 excellent
**Presentation:** 4 excellent
**Contribution:** 3 good
**Rating:** 8
**Confidence:** 4

**Summary:**

COPL (Clip-guided Open vocabulary Policy Learning), the proposed algorithm in this paper, attempts to solve the problem of open vocabulary language instructed reinforcement learning (RL) -- the task of accomplishing a goal described in natural language without having any constraints (ideally) on the words used to specify the instruction. The paper works in a setting where the behaviors are still fixed, for instance, the agent still has to perform a similar sort of sequence of actions like hunting, but the object of hunting is chosen from objects that did not occur during training and are rather chosen from other available objects that can be hunted. COPL works as follows: first, the object that is to be acted on is segmented out using a modified version of MineCLIP. This gives a confidence map. This confidence map, combined with a focal objective that tries to get the desired object in the center of the frame and nearer to the agent, forms what is called as focal reward. The agent looks at the current observation from the environment in addition to the confidence map derived and is asked to take action. These actions are then optimized using standard RL algorithms such as PPO on a combination of focal reward with reward coming from the environment.

The experimental analysis involves (i) showing how the proposed (selective segmentation + focus)  works on a single task, e.g., only hunting a pig, (ii) working of the algorithm on multi-task settings, e.g., hunting a pig and hunting a sheep (plus, cow and chicken), and (iii) open-vocabulary testing of agent's capabilities. The approach is compared with different reward combinations and ablations of the proposed reward.

**Strengths:**

Writing and presentation quality is very high. The paper is dense with content and I enjoyed going through the paper multiple times. The architecture description is neat, although I had to assume few things about the implementation while evaluating the correctness.

The sections are introduced in a logical order, and every choice behind the focal reward is well-motivated. As I understand it, the focal objective is very similar to how a human would accomplish the task of, say, 'hunting a pig', by being confident about the animal that is to be approached and getting a better focus. This description might seem to overfit the case of hunting, but other tasks that require inferring the intended object correctly and approaching it efficiently are also covered. This also stems from other problems, which I discuss in the weakness section.

The approach described in the paper is compared with relevant baselines while comparing single-task, multi-task, and open vocabulary capabilities.

More importantly, I find this work important from the point of view of starting a discussion on how open-vocabulary instructions can be used in conjugation with RL. The overall methodology and evaluation framework outlined is quite systematic and serves as a guide for future research in this area.

**Weaknesses:**

**Limits of confident and focused seeking of object**: From my limited knowledge of Minecraft as a game, it is an open-ended environment where the agent can build by gathering resources and surviving. It is open-ended in the sense that one gets to express complex ideas, which involves using resources through intents. I am not sure all this open-endedness is captured in being confident about the desired object and focusedly approaching it. Put another way, the approach might be overfitting only a part of the actual space of possible Minecraft behaviors.

**Issue with negative words**: For segmenting out the object in the intent, the method uses negative words. While this approach would work in domains such as Minecraft, where the entities are finite and known, I am unsure whether it will hold when applied to real-world cases where entities could be unknown and infinite.

**Comment on novelty**: I am not fairly acquainted with Minecraft research. From the related works pointed out in the paper, it does seem that the approach presented is novel in its entirety. But, from the computer vision perspective, both prompt-based local segmentation and focal vision are pretty standard. The use of CLIP for Minecraft is the prior work that COPL builds on. So, I find the novelty of COPL in applying everything in a functional manner to test the open-vocabulary capabilities of the assembled system.

**Questions:**

I have the following questions for the authors:
1. By limiting the CLIP model to a set of pre-determined negative words, isn't the paper limiting the scope and moving away from the actual aim of being open vocabulary?
2. To extend the previous question, is it possible to perform a similar analysis, but instead of negative words, use the entire vocabulary?
3. Would it make sense to keep the objects fixed and change the intended behavior to a similar but nuanced variant of the behavior? Again, I have limited knowledge of Minecraft as a game and have limited knowledge about possible behaviors. However, it seems very logical to me to test open vocabulary capabilities where the agent might 'catch' an animal rather than 'kill' an animal where catching is out of distribution. These behaviors are not very different from training behavior as 'exploring the world' is to it.

---

> ### Author Response · Authors · 2023-11-15
>
> Thank you for your efforts and valuable comments. We appreciate such a detailed and insightful discussion on our method. We would like to address the weaknesses and questions you raised in the review.
>
> > By limiting the CLIP model to a set of pre-determined negative words, isn't the paper limiting the scope and moving away from the actual aim of being open vocabulary?
>
> These pre-determined negative words are used to instruct the model on what to ignore, thereby enhancing the accuracy of the 2D confidence map by reducing noise. Concretely, we compute similarities between these negative words and image patches, in addition to similarities between the target object name and image patches. By applying a softmax function to these similarities on each patch, we suppress the activation of the target object on non-target patches, especially where objects from the negative word list are likely present. This results in a cleaner, low-noise confidence map.
>
> Given the principle mentioned above, we construct the negative word list containing objects that frequently appear in Minecraft for effective denoising. Note that while we limit the negative words, we do not restrict the range of target objects. So in principle, our modified MineCLIP can generate the confidence map for any target object and does not move away from the aim of being open-vocabulary.
>
> > To extend the previous question, is it possible to perform a similar analysis, but instead of negative words, use the entire vocabulary?
>
> If 'the entire vocabulary' refers to all object names in Minecraft, we believe it will work as well because our negative is equivalent to the most frequent part of these objects and is proven to be effective. If 'the entire vocabulary' means the general vocabulary that CLIP can understand, we suppose that it would be less effective. There are multiple reasons. For example, synonyms for the target object will suppress its activation on the confidence map because the synonyms also have high similarities on the patches where the target object presents. Additionally, words with different levels of granularity will complicate the segmentation results. For instance, a human might be identified as 'human' as a whole but also as 'head', 'arms', etc., separately.
>
> Therefore, a pre-determined, domain-specific word list is essential for open-vocabulary tasks. This is also evidenced in open-vocabulary learning studies, such as open-vocabulary segmentation [1,2,3], where benchmark datasets would provide a pre-determined class set for segmentation evaluation. We hope this could also address your concerns about negative words raised in Weaknesses.
>
> [1] Ding et al., 2022, Open-Vocabulary Panoptic Segmentation with MaskCLIP.
>
> [2] Zhou et al., 2022, Extract Free Dense Labels from CLIP.
>
> [3] Liang et al., 2023, Open-Vocabulary Semantic Segmentation with Mask-adapted CLIP.
>
> > Would it make sense to keep the objects fixed and change the intended behavior to a similar but nuanced variant of the behavior?
>
> We appreciate this insightful suggestion and conduct relevant tests to explore our model's ability to adapt to similar but nuanced behavior patterns. We test the hunt domain model on the 'milk a cow' task, where the object 'cow' is familiar, but the required behavior pattern is slightly different. In detail, hunt tasks require 'attack' action to interact but 'milk a cow' requires 'use' action. The hunt model achieves a 43(±16)% success rate, lower than that of the harvest model trained on this task but significantly higher than zero, suggesting that COPL can partially generalize to a nuanced variant of the training behavior pattern.
>
> Another relevant example is the test task 'harvest sand' in the harvest domain, where both object and behavior are out of the training distribution. As we introduced at the end of Appendix B.3.3, 'harvest sand' requires a slightly different behavior from the training task, and at the same time, 'sand' is a novel target object. As demonstrated in Figure 8(c), COPL outperforms the baseline EmbCLIP on this task.
>
> However, it is important to note that this generalization may stem from the stochastic or noisy nature of the policy output and does not represent that COPL has a true open-vocabulary ability over verbs.
>
> ----
>
> If you have any additional questions or comments, please do not hesitate to let us know. We are more than happy to provide any further clarifications or information that may be helpful.

---

> > ### Comment · Reviewer_9Xm8 · 2023-11-21
> >
> > Thanks for your detailed response. My questions are answered to a good extent. I do not have anything more to add.

---

### Official Review · Reviewer_6CKq · 2023-11-01

**Soundness:** 3 good
**Presentation:** 3 good
**Contribution:** 2 fair
**Rating:** 5
**Confidence:** 5

**Summary:**

The paper proposes a new intrinsic reward for open-vocabulary tasks in minecraft. The proposed technique first applies existing dense CLIP methods to the MineCLIP model and discovers similar open-vocabulary segmentation property. The authors then proposes an intrinsics reward that motivates the agent to approach the segmented object. With this intrinsics reward, the paper allows the minecraft agent to learn to perform certain open-vocabulary tasks. The authors demonstrate the effectiveness of the method in terms of single tasks, multi-tasks, and open-vocabulary setting.

**Strengths:**

- The writing of the paper is clear. Method and motivations are discussed thoroughly.
- The paper introduces spatial priors for its intrinsics reward which was missing in vanilla way of using MineCLIP
- Evaluation is thorough despite limitations I will mention in the weakness section.

**Weaknesses:**

While I acknowledge the soundness of the approach and the presentation, I don't find this paper's contribution significant enough for acceptance. This is the main reason of my rating.

The core contribution of the paper is an intrinsics reward for minecraft with a lot of limitations. 1. the reward seems to be specifically tailored for minecraft's first-person-view setting, and specifically towards tasks that involves approaching an object 2. the tasks have to be object-centric, and the generalization is mostly object level.

While the promise of this paper / minecraft itself is about open-vocabulary, the presented method is limited to approaching objects in fpv setting. This is also reflected in the evaluation, where the task covered are no way near open-world. This also has been mentioned by multiple other reviewers. The authors should tune down their claim about open-world.

The proposed open-vocab segmentation seems to be very similar to previous methods like [1]. Even if I disregard this fact, it seems to me that if this reward is already tailored for minecraft (first-person view + task is mainly approach object), one may well use some open-vocabulary detection model tailored for the few tasks the paper benchmarked in. Once one detects the objects from text, the reward the authors propose seems an obvious thing to do. The evaluation, as a result of the limitations of such reward, are also constrained to be very object centric ones, which breaks the purpose of MineCLIP. I would not claim the proposed technique to be effective for general open-vocabulary tasks.

After the rebuttal, I decide to lift my score from 3 to 5 for added result and also raise my confidence from 4 to 5. This is because I had personal experience trying almost every single component the paper used, and have tried some them on the figures the authors provided during rebuttal period. I believe the current approach have its merit, but would belong to a more system/experiment heavy paper where such a reward only plays part of the role. At its current state, I reiterate my belief that such a reward alone, under the broken promise of open-worldness, doesn't constitute the technical contribution a full ICLR paper needs.

[1] https://arxiv.org/pdf/2112.01071.pdf

**Questions:**

1. I am not exactly sure whether the authors claim the architecture in figure 2 to be a main contribution. If so, the authors should probably discuss previous approaches and how is your approach different, either when you mention MaskCLIP for the first time or in related work.

2. From my understanding, for this reward to work properly, the object has to be already in FOV, correct?

---

> ### Author Response · Authors · 2023-11-15
>
> Thank you for your efforts and valuable comments. We would like to address the weaknesses and questions you raised in the review.
>
> > I am not exactly sure whether the authors claim the architecture in figure 2 to be a main contribution.
>
> The architecture illustrated in Figure 2 is not the main contribution of our study but rather a key module within our overall framework. As introduced in Section 3.1, it is adapted from MaskCLIP. The differences between our modified MineCLIP and MaskCLIP are twofold: firstly, MineCLIP includes an additional temporal transformer in the vision pathway; secondly, our modified MineCLIP outputs a confidence map for the target object instead of a semantic segmentation result.
>
> > From my understanding, for this reward to work properly, the object has to be already in FOV, correct?
>
> Getting a non-zero focal reward indeed requires the target object to be in the agent's field of view. When the target object is not present in the FOV, resulting in a confidence map with no high-probability patches, the focal reward is zero. This characteristic would prompt the agent to look around immediately after spawning in the Minecraft world to find the target object. In our experiments, target objects often appear in the agent's observable range, which means that the agent can see the target object when facing the direction of the target object.
>
> If we consider a more challenging setting where the target object is located out of the agent's observable range at the spawning point, an additional exploration algorithm should be integrated to encourage the agent to move around. For example, incorporating novelty-induced exploration.
>
> > The core contribution of the paper is an intrinsics reward for minecraft with a lot of limitations ...
>
> We acknowledge that our method is primarily designed for first-person view and object-centric tasks. But we want to clarify that this setting is not exclusive to Minecraft, but contains a wide range of realistic and general applications. For example, embodied AI research domains, such as autonomous driving, navigation, and robotic manipulation, often employ first-person views and/or focus on object-centric interactions. This setting better mirrors the complexity of real-world challenges compared to canonical RL environments, where our method may not be suitable.
>
> Additionally, we would like to emphasize that our contribution also includes a novel policy conditioning representation, namely the 2D confidence map. Using a 2D confidence map to replace language input substantially improves the generalization ability of learned policy, as demonstrated in Section 4.2. Although currently restricted to object-centric tasks, it offers some insights about grounding language into vision to facilitate multimodal policy learning.
>
> We hope this elaboration clarifies the generality and contributions of our work, while also acknowledging and appreciating the limitations you have pointed out.
>
> > ... which breaks the purpose of MineCLIP.
>
> As mentioned above, our method is primarily designed for first-person view and object-centric tasks, which are not restricted in Minecraft. We use Minecraft as a convenient and suitable testbed to validate our method and adapt MineCLIP to provide the segmentation functionality that our method requires. Therefore, we do not intend to develop a new fundamental model or method that follows the purpose of MineCLIP and theoretically surpasses it in terms of multimodal ability and generality in Minecraft. Instead, it just serves as a key module specific to Minecraft within our framework to solve the problem we defined in Section 2. In other first-person view and object-centric environments, our framework can adapt by replacing the modified MineCLIP with general models like Grounding DINO + SAM or domain-specific models.
>
> ---
>
> If you have any additional questions or comments, please do not hesitate to let us know. We are more than happy to provide any further clarifications or information that may be helpful. If you find our responses satisfactory, we would be grateful if you could raise the score.

---

> > ### Comment · Reviewer_6CKq · 2023-11-19
> >
> > Dear authors, thank you for your quick response. Before I make my comprehensive response to your rebuttal, I'd really appreciate it if you can give a convincing response to my point about open-vocabulary detection model under the same adoption to minecraft(or fpv games). I deem a clear explanation and supporting experiments to be essential to address that point.

---

> ### Author Response · Authors · 2023-11-19
>
> Dear reviewer, we appreciate the opportunity to further discuss and respond to your point about the use of open-vocabulary detection models in Minecraft or other first-person-view (FPV) games.
>
> **Minecraft**. Firstly, we would like to clarify that currently, there is not an existing open-vocabulary detection model tailored for Minecraft. An alternative approach is to use some off-the-shelf open-vocabulary object detection models directly. However, as detailed in our newly appended Appendix A.4, due to the significant domain gap between visuals in Minecraft and the real world, direct application of these models to Minecraft results in inaccurate detection. In other words, they require adaptation to the Minecraft domain, implying additional costs associated with labeled data collection and fine-tuning. In contrast, our modification on MineCLIP does not demand further fine-tuning, allowing for immediate and effective implementation.
>
> **Other FPV games.** Similar to Minecraft, other FPV games also suffer from the domain gap between their visual styles and the real world more or less. As mentioned previously, if we want to adapt an off-the-shelf open-vocabulary detection model to a specific game domain, a labeled game dataset is required. Annotating object bounding boxes and names demands extensive human labor. However, the wealth of game videos on the Internet, such as those on YouTube, presents an alternative approach. These videos often come with subtitles, providing natural labels that can be utilized for training. Therefore, compared to adapting an object detection model, training a CLIP-like vision-language model (VLM), such as CLIP [1] and CLIP4Clip [2], on this massive data would be much more labor-saving and automated. Most importantly, MineCLIP has already demonstrated this route is feasible. Once a domain-specific CLIP-like model is obtained for a game, our method could be then implemented based on this VLM to guide the agent to learn basic skills in this game.
>
> In summary, compared to tailored open-vocabulary detection models, our method could be nested in a general pipeline with VLM learning, which requires less labor and is more automated, for FPV games. We hope the discussion above addresses your concern. If you still have any additional questions or comments, please do not hesitate to let us know.
>
> [1] Radford et al., 2021, Learning Transferable Visual Models From Natural Language Supervision.
>
> [2] Luo et al., 2021, CLIP4Clip: An Empirical Study of CLIP for End to End Video Clip Retrieval.

---

> ### Comment · Reviewer_6CKq · 2023-11-19
>
> Dear authors,
>
> > However, as detailed in our newly appended Appendix A.4, due to the significant domain gap between visuals in Minecraft and the real world, direct application of these models to Minecraft results in inaccurate detection.
>
> This is not True, because open-vocabulary detection easily generalizes in many video games just as well as CLIP (in fact many of them use CLIP). A while ago I tried [grounded-SAM](https://github.com/IDEA-Research/Grounded-Segment-Anything) on minecraft. Its open-vocabulary detection worked perfectly. I just tried a few images from you paper's Figure 3, and it worked perfectly with a fast speed. The model successfully detected sheep, flower, cow, even the sword.
>
> In addition, I personally tried every single figure in your Appendix figure 9, following one negative word of "grass" which you also did in method. Grounded DINO was able to detect the object 6/6. In your added figure 11, grounded dino got 4/6 correct from your result, but the original prompt I tried is "minecraft sheep" for (d) and it succeeded, so I'd consider open-vocabulary detection to work in 11/12 cases. I believe you can potentially get better results with higher resolution screenshots. I think such result proves they are strong replacements even without minecraft specific thing.
>
> To addresses my concerns, I believe results with grounded SAM for example, or results in other FPV domains are necessary, which can be use to defend the merit of your method.

---

> > ### Author Response · Authors · 2023-11-21
> >
> > Dear reviewer, thank you for your response.
> >
> > Considering that Grounded SAM is essentially Grounding DINO + SAM, where Grounding DINO offers bounding boxes of target objects as prompt to SAM, we only discuss Grounding DINO here. We believe that the effectiveness of Grounded SAM can be inferred from that of Grounding DINO.
> >
> > > In your added figure 11, grounded dino got 4/6 correct from your result ...
> >
> > We would like to clarify that Grounding DINO actually got 3/6 correct as shown in Figure 11: (b) misidentifies cow as pig; (d) fails to detect sheep; (f) misidentifies horse as llama.
> >
> > > The model successfully detected sheep, flower, cow, even the sword.
> >
> > To offer a comprehensive understanding of the effectiveness of Grounding DINO, we newly added Appendix A.5 in the revision. As shown in Figure 12, we find that Grounding DINO consistently identifies the real cow, horse, and llama as distinct objects, when provided different targets for detection. This observation suggests that Grounding DINO **is capable of generating these bounding boxes probably based on low-level features but cannot accurately match their semantic meanings in the Minecraft domain with target names**. In contrast, our modified MineCLIP exhibits clear distinctions between these animals, as demonstrated in Figure 13. Moreover, the high accuracy of COPL on unseen instructions (~90%) serves as additional evidence of our method's efficacy, as shown in Figure 7(c).
> >
> > > ... following one negative word of "grass" which you also did in method ...
> >
> > This use of negative words is a good inspiration to enhance the ability of Grounding DINO. Therefore, we also construct a word list containing cow, sheep, pig, chicken, horse, llama, spider, mushroom cow, flower, tree, sword, and grass. These words encompass most objects in our evaluation images. The results shown in Figure 15 suggest that even given a word list, Grounding DINO still cannot detect objects accurately. Specifically, it fails to detect pig (first and fifth images), sheep (second image), or chicken (third image). Also, it misidentifies llama (fouth image), cow (ffith image), and mushroom cow (last image).
> >
> > We speculated that the background word 'grass' might be too strong and thus suppress the activation of other objects. Therefore, we remove 'grass' from the word list and re-evaluate the model. The result shown in Figure 16 is also not satisfied, as most problems observed in Figure 15 persist.
> >
> > > but the original prompt I tried is "minecraft sheep" for (d) and it succeeded
> >
> > To further explore the potential impact of domain-specific prompts, we experimented by adding the prefix "minecraft" to each word in the word list. As reported in Figure 17, we find that such a “domain prompt” does not yield improved detection results either. What is worse, we observe a more severe missing detection in the last two images.
> >
> > By these exhaustive evaluations, we show that the performance of Grounding DINO is notably limited by the domain gap between the real world and Minecraft. The model exhibits **significant misidentification and missing detection** in Minecraft domain.
> >
> > > I believe you can potentially get better results with higher resolution screenshots
> >
> > For a fair comparison, we provide both Grounding DINO and MineCLIP with images in the size of 160 \* 256.
> >
> > > To addresses my concerns, I believe results with grounded SAM for example, or results in other FPV domains are necessary, which can be use to defend the merit of your method.
> >
> > We are running an experiment where rewards are calculated using the segmentation map generated by Grounded SAM. However, the introduction of Grounded SAM involves a large amount of additional computation, resulting in a much slower training speed compared to our original method. Our modified MineCLIP can provide segmentation results **at a faster pace and more lightweight computation**.
> >
> > We will update our results as soon as this experiment is finished.

---

> > > ### Comment · Reviewer_6CKq · 2023-11-21
> > >
> > > Hi authors,
> > >
> > > In case you don't have time to finish my requested additional experiments, there is another simpler thing that will help understanding a lot. Since we have contradicting experience with open-vocabulary detectors, it would be helpful to do some unbiased test about it, which will also enhance the evaluation of the paper.
> > >
> > > You can conduct such an experiment with your method vs Grounded SAM detection and list the result side by side.
> > > For data, take object from [pig, cow, sheep, mushrooom cow, tree, flower, horse, torch], google search "minecraft [object name] screenshot" in image tab, and take the first two images that includes the item and the has item fully in field of view.
> > > This boils down to a total of 16 test images (although ideally you paper should have even more diverse examples)
> > >
> > > Run Grounded-SAM's huggingface demo with just one negative class "grass".
> > > You can run your method to get bounding box with whatever negative classes you want, but the set of negative words have to remain the same across all images.
> > >
> > > I believe such a test would be a fairer quantitative comparison.

---

> > > > ### Author Response · Authors · 2023-11-22
> > > >
> > > > Dear reviewer,
> > > >
> > > > We follow your requirement to obtain images of pig, cow, sheep, mushroom cow, tree, flower, horse, and torch. The results are shown in the newly added Figure 25 and Table 9. Since our method is essentially segmentation instead of object detection, we cannot generate bounding boxes for target objects. Instead, we assess the success of our method by examining whether the activation of patches containing the object is significantly higher than that of the background. The results demonstrate that Grounded SAM successfully detects 15 of 28 objects in these images, while our method successfully detects 26 of 28 objects.
> > > >
> > > > We hope this fairer quantitative comparison can address your concern.

---

> > > > > ### Comment · Reviewer_6CKq · 2023-11-22
> > > > >
> > > > > Dear author,
> > > > >
> > > > > This is good but could you follow my exact instruction for grounding-DINO? aka using only [cow, grass] for an image with cow, instead of a whole list of objects. I think this is a fairer comparison as many previous method already use LLM to derive object names to detect from instruction, and such as [this paper from 2022](https://nlmap-saycan.github.io/)
> > > > >
> > > > > I will consider raising my score after this is done as well as reading other reviewer's updates.

---

> > > > > > ### Author Response · Authors · 2023-11-22
> > > > > >
> > > > > > Dear reviewer,
> > > > > >
> > > > > > We would like to clarify that we indeed use only [object name, grass] for an image with the corresponding object. This is evidenced by the tree, an easily detected object for Grounded SAM, which does not have bounding boxes in other images.

---

> > > > > > ### Author Response · Authors · 2023-11-23
> > > > > >
> > > > > > Dear reviewer,
> > > > > >
> > > > > > We are happy to share that we have finished the experiment where rewards are calculated using the segmentation map generated by Grounded SAM. Considering the limitation of rebuttal time, we chose a single task 'shear a sheep' for evaluation. The results are shown in Figure 26. We observe that the performance of Grounded SAM reward is similar to that of sparse reward, implying that the segmentation map generated by Grounded SAM might be noisy and could not provide accurate location information of about target object 'sheep'.
> > > > > >
> > > > > > We really appreciate your effort and patience in reviewing our paper and detailed discussion with us.

---

> ### Comment · Reviewer_6CKq · 2023-11-23
>
> Here is my final review after taking a look at other reviews' comments. I've edited response, score and confidence to reflect my final opinion.
>
> I appreciate the authors response to my questions about open-vocabulary detector. The additional ablations contribute to my score increase. However, as all other reviewers pointed out, the paper's method has a lot of limitation that contradicts open-world promise of the claim.
>
> While the promise of this paper / minecraft itself is about open-vocabulary, the presented method is limited to approaching objects in fpv setting. This is also reflected in the evaluation, where the task covered are no way near open-world. This also has been mentioned by multiple other reviewers. The authors should tune down their claim about open-world.
>
> After the rebuttal, I decide to lift my score from 3 to 5 for added result and also raise my confidence from 4 to 5. This confidence score is due to the fact that I had personal experience trying many components the paper used before ICLR (Dense CLIP, MineCLIP, inverse bbox size for distance, LLM extract objects from task, minecraft, open-vocab detection on minecraft), and have tried one of them on the figures the authors provided during rebuttal period. Such confidence score raise has no conflict of interest, as I don't have any in-submission or on-going paper that's close enough to the author's.
>
> I acknowledge current approach's merit, but would believe it has to belong to a more system/experiment heavy paper where such a reward only plays part of the role. Many many more diverse tasks, or a whole end-to-end decision making system would make this paper much stronger. At its current state, I reiterate my belief that such a reward design alone, under the broken promise of open-worldness, doesn't constitute the technical contribution a full ICLR paper needs. On the other hand, I think the paper is above a score of 3 once the authors would incorporate all our feedbacks. If my confidence score between 4 and 5 is an absolute tie breaker for the AC, I recommend accept this paper.

---

### Official Review · Reviewer_Q4te · 2023-11-01

**Soundness:** 3 good
**Presentation:** 3 good
**Contribution:** 3 good
**Rating:** 6
**Confidence:** 3

**Summary:**

This paper presents a method for training an open-vocabulary policy on Minecraft tasks via RL guided by CLIP. Intrinsic rewards for the RL policy are computed based on the patch-wise CLIP similarity between the mentioned target object in a given language instruction and the image of the enviornment. Confidence maps based on the patch-wise similarity are passed into the policy in addition to visual information. The choices for the intrinsic rewards and inputs to the policy allow for some invariance in behavior to be learned across target objects, permitting some generalization when performing a seen task on an unseen object.

**Strengths:**

- Overall, the paper is clear and easy to read.
- The paper has some interesting insights about using CLIP guidance in the Minecraft context -- including the focal reward, the subtraction the probabilities of negative classes as a denoising procedure, and the way in which the patch similarities/probabilities are determined from MineCLIP.
- The paper includes ample architecture and implementation details in the Appendix which promotes reproducibility.

**Weaknesses:**

- The broad idea of using VLMs to allow for open-vocabulary manipulation with objects has been explored previously (e.g. MOO, Stone et al. 2023), though the paper does have some interesting insights about applying VLM guidance that are particular to the Minecraft setting, as mentioned above.
- As is common with many reward shaping approaches, the hyperparameter $\lambda$ must be tuned to determine the weighting on the focal reward. According to Figure 11 in the Appendix, the choice of this hyperparameter can have a significant effect on the results. Is the same value of $\lambda$ optimal across multiple task families?
- The approach does have some hand-crafted components that are specific to this domain. For example, the Gaussian kernel in the focal reward relies on the fact that interaction in Minecraft occurs "when the cursor in the center of the agent view aligns with the target" and also guides the agent to focus on a single target rather than multiple. How was this kernel chosen and how sensitive is the performance of the method to the specific choice of kernel? Another example is the negative word list; while effective for denoising, it is determined in a domain-specific fashion, so it is unclear if the benefit of this procedure would be helpful for other domains.
- The choice of adding the unified 2D confidence map to the policy input is an interesting way to get some invariance across objects. But removing the natural language input constrains the policies to be single task instead of multi-task policies. What is the advantage of removing natural language? One rationale might be that unseen objects which are OOD for the policy do not have to be encoded by the text encoder--but these unseen objects are already being encoded by the visual encoder, so it is unclear if this is the reason. Was the choice of not including text as a policy input ablated?

**Questions:**

- How sensitive is $\lambda$ across multiple task families?
- How was the Gaussian kernel constructed?
- Why were natural language instructions not included as an input to the policy?
- Given that the language instructions are fairly simple, what is the rationale for using an LLM to find the target object?

---

> ### Author Response · Authors · 2023-11-15
>
> Thank you for your efforts and valuable comments. We would like to address the weaknesses and questions you raised in the review.
>
> > Another example is the negative word list; while effective for denoising, it is determined in a domain-specific fashion, so it is unclear if the benefit of this procedure would be helpful for other domains.
>
> We would like to clarify that constructing a negative word list is inherently domain-specific. For instance, in another domain, indoor navigation tasks, the list should be constructed to contain indoor objects rather than outdoor ones. Although the approach may vary with each domain, the concept of building a task-relevant negative word list to denoise and enhance segmentation effectiveness is universally applicable.
>
> > How sensitive is $\lambda$ across multiple task families? Is the same value of $\lambda$ optimal across multiple task families?
>
> In our experiments, we observe that the average focal reward per step consistently falls into a range of 0.1 to 0.3 across all tasks due to the similar state spaces (all tasks are evaluated in Minecraft). The focal rewards' uniformity in order of magnitude across task families indicates the robustness of our $\lambda$ selection, although it is derived from 'hunt a cow' task and 'hunt a sheep' task. To validate this, we conduct additional experiments on two tasks from the harvest task family, 'milk a cow' and 'shear a sheep', and add the results to Appendix B.4. As shown in Figure 20(a) and (b), $\lambda=5$ still outperforms other values on these two tasks.
>
> While we acknowledge that $\lambda=5$ may not be the exact optimal value for every individual task, it is a reasonably near-optimal choice (*e.g.* on the same order of magnitude as the optimal value), serving as a broadly effective parameter for our multi-task training.
>
> > How was the Gaussian kernel constructed? How sensitive is the performance of the method to the specific choice of kernel?
>
> The Gaussian kernel in our method is constructed based on the simple principle that values should be higher near the center and decrease with distance. $\sigma=(H/3,W/3)$ is an intuitive choice, aiming to maintain high contrast between the center and the edge, as well as between the edge and areas outside the field of view.
>
> To quantitatively evaluate this intuition and answer the second question, we conduct additional experiments to test the sensitivity of our methods to Gaussian kernels with different variances and add the results to Appendix B.4. The experimental settings are the same as those in Figure 6(c) and (d). As illustrated in Figure 21, $\sigma=(H/3,W/3)$ outperforms the other two choices. We suppose that the Gaussian kernel with $\sigma=(H/2,W/2)$ is too wide to provide sufficient contrast between the center and the edge, while the Gaussian kernel with $\sigma=(H/5,W/5)$ is too narrow to provide sufficient contrast between the edge and areas outside the field of view.
>
> > Why were natural language instructions not included as an input to the policy?
>
> We do not include the natural language instruction as input to the policy primarily because the instruction cannot offer additional useful information beyond what is already provided by the 2D confidence map in our Minecraft experiments. The information in the language instruction, besides the target object, often consists of verbs like 'hunt', 'harvest', or 'collect'. However, such actionable information does not significantly aid in task completion. On the one hand, the action patterns required to complete tasks in Minecraft are relatively simple, *i.e.*, move towards the target object and take 'attack' or 'use' action. On the other hand, the mapping between verbs and action patterns is not one-to-one. For example, an instruction with the verb 'collect' may require the agent to either 'attack' or 'use' depending on the type of the target object. Therefore, we believe that language instruction cannot provide additional useful information for the agent's policy.
>
> We acknowledge that in other domains like robotic manipulation, language instructions often contain essential information, such as verbs 'pick' and 'push' indicating totally different behavior patterns. Under this circumstance, taking as input the language instruction will be necessary.

---

> ### Author Response · Authors · 2023-11-15
>
> > Given that the language instructions are fairly simple, what is the rationale for using an LLM to find the target object?
>
> The use of an LLM is intended to address the challenge where the name of the target object is not explicitly mentioned in the language instruction. For example, consider the instruction 'obtain wool with shears'. Here, 'wool' is the target item to acquire, but the actual target object for the agent to approach and interact with is 'sheep', which does not appear in the instruction. It is hard to design a programmatical method to extract such an implicit target object from the instruction. Consequently, we resort to LLMs, which understand the underlying relationship between the target item (wool) and the target object (sheep). More examples are available in Appendix A.1.
>
> ----------
>
> If you have any additional questions or comments, please do not hesitate to let us know. We are more than happy to provide any further clarifications or information that may be helpful. If you find our responses satisfactory, we would be grateful if you could consider raising the score.

---

> > ### Comment · Reviewer_Q4te · 2023-11-21
> > **Response to author**
> >
> > Thank you for the response and the clarifications. The approach has some hand-crafted components that are specific to this domain, and further, is restricted to tasks that involve approaching an object, and is a bit sensitive to tuning. As other reviewers have mentioned, the contribution is somewhat limited in scope. I maintain my score of weak accept and will continue to monitor the discussions.

---

### Meta-Review · Area_Chair_dFC4 · 2023-12-07

**Metareview:**

This paper proposes a method for training agents to complete object-focused tasks defined by natural language in an open world game (Minecraft). The framing of the initial paper submitted caused some confusion over the type of tasks the method is capable of due to emphasis on the open world environment. Whilst this was mostly resolved through discussion and revisions to the paper, concerns remained among some reviewers about the soundness of the paper's conclusions based on prior experience with related approaches. Whilst the approach appears to show early promising results, further work could improve readers' confidence in the precise scope of applicability and conclusions reached regarding existing methods.

**Justification For Why Not Higher Score:**

+ Too many open concerns from multiple reviewers at the end of the review period

**Justification For Why Not Lower Score:**

N/A

---

### Decision · Program_Chairs · 2024-01-16

Reject